# EXPANDING THE ACTION SPACE OF LLMS TO REASON BEYOND LANGUAGE

## ABSTRACT

Large Language Models (LLMs) are powerful reasoners in natural language, but their actions are typically confined to outputting vocabulary tokens. As a result, interactions with external environments—such as symbolic operators or simulators—must be expressed through text in predefined formats, parsed, and routed to external interfaces. This overloads the model's language with both reasoning and control duties, and requires a hand-crafted parser, external to the LLM. To address this, we decouple environment interactions from language by internalizing them in an Expanded Action space (ExpA), beyond the vocabulary. The model starts reasoning in the default language environment, but may trigger routing actions and switch to an external environment at any time. From there, the model can only invoke environment-specific actions, receive feedback from the environment, and potentially route back to language as a result. To promote effective exploration of the expanded action space and new environments, we introduce ExpA Reinforcement Learning (EARL) with counterfactual policy optimization. On tasks requiring multi-turn interactions and contingent planning, EARL outperforms strong baselines with vocabulary-constrained actions. It performs robustly across calculator-based multi-task learning and, in the partially observed sorting problem, achieves perfect Sort-4 accuracy while self-discovering an efficient algorithm competitive with classical designs.

## 1 INTRODUCTION

Recent advancements in Large Language Models (LLMs) have extended their role from pure language reasoners to versatile agents capable of interacting with external environments, including tools, APIs, and embodied systems (Shao et al., 2024; DeepSeek-AI et al., 2025; Qin et al., 2024). This development is motivated by two complementary perspectives. First, external environments can augment LLMs by providing capabilities they lack inherently, such as exact symbolic computation (Lee et al., 2024) or access to up-to-date knowledge (Schick et al., 2023). Second, LLMs can extend their reasoning into external environments by mapping language instructions into operations such as API calls or robotic control, allowing them to solve tasks in the digital or physical world (Qin et al., 2024; Li et al., 2023; Szot et al., 2025; Xiang et al., 2023).

LLMs from previous works in this area can be viewed as agents acting in decision processes with an action space restricted to vocabulary tokens $\mathcal{V}$, as illustrated in Figure 1a. The agents operate only in natural language, selecting to-

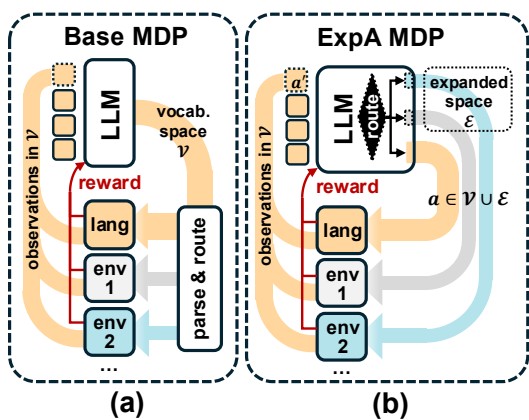

Figure 1: The Markov Decision Process (MDP) of LLM interacting with external environments. (a) In existing works, LLM is confined to its vocabulary space $\mathcal{V}$ for both reasoning and interaction with external environments, where the latter requires an external parser to detect special patterns. (b) We decouple environment interactions from language by internalizing them as an Expanded Action space (ExpA) $\mathcal{E}$ beyond vocabulary.

kens to append to an observation sequence. Interactions with external environments are mediated through a parser, which translates predefined text patterns (*e.g.*, tool tags or structured JSON) into environment-specific actions, routed to the corresponding environment (Schick et al., 2023). The environment executes the actions and returns a plain-text observation in $\mathcal{V}$, which is appended to the model's context. To teach models to adopt such interactions, existing works use in-context examples, and often additional training, such as supervised fine-tuning on labeled tool calls (Schick et al., 2023; Luo et al., 2025), or reinforcement learning (RL) (Feng et al., 2025) with rewards determined by language outputs (Feng et al., 2025; Singh et al., 2025) or by the environments (Qin et al., 2024).

We propose a fundamental shift from the language-only paradigm. Our aim is threefold: (1) to decouple environment interactions from language reasoning, (2) to enable end-to-end training by removing reliance on external parsers and keeping interactions under the model's control, (3) to fully support RL on base models, *i.e.*, Zero-RL (DeepSeek-AI et al., 2025), without requiring supervised tool-call data or adherence to predefined language patterns. Our solution is outlined below.

We introduce an Expanded Action space $\mathcal{E}$ (**ExpA**) that extends models' capabilities beyond outputting vocabulary tokens by creating actions for direct interaction with external environments. In the default language environment, the model can either reason by generating tokens from $\mathcal{V}$ or trigger a routing action $g_i \in \mathcal{E}$ to activate a specific environment $i$ (*e.g.*, a calculator), appending a predefined description (*e.g.*, "calculate") to the sequence. Once environment $i$ is active, the model is restricted to a set of environment-specific actions in $\mathcal{E}$ (*e.g.*, calculator buttons), which yield observations in $\mathcal{V}$ (*e.g.*, pressed buttons or calculation results), and upon completion (*e.g.*, pressing "="), return control to the language environment. As illustrated in Figure 1b, this paradigm achieves a clean separation between language-based reasoning and environment interaction. Importantly, ExpA is fundamentally distinct from simply expanding the *token space*, as is common in multimodal LLMs Chen et al. (2025b); Wang et al. (2025a). Since external actions are not used as model inputs, ExpA avoids the need for costly fine-tuning to represent new actions tokens in the LLM's input, enabling more efficient and modular integration of environment-specific actions.

A key challenge when introducing LLM agents to external environments is that the pre-trained models lack experience acting in and observing them. When expanding the action space, introducing new model parameters, there is no guarantee that the agents will interact with the new environments to solve problems. To address this, we employ ExpA with RL (**EARL**), introducing a novel counterfactual policy optimization strategy to encourage exploration of new environments. During training, for each rollout, we construct a counterfactual trajectory by forcing a routing action at a plausible intermediate step, identified as a position where the model assigns high probability to the routing description token. The advantage is then computed as the difference between the counterfactual and original rewards, thereby encouraging exploration of rarely visited but essential environments.

In summary, we establish a principled and scalable framework for reasoning beyond language with the following contributions:

- **Expanded Action space (ExpA):** a new paradigm that decouples language reasoning from environment interaction by introducing explicit routing and environment-specific actions.
- **ExpA Reinforcement Learning (EARL):** an algorithm based on counterfactual rollouts that encourages exploration of rarely invoked but crucial environment interactions.
- **Implementation:** efficient support for ExpA rollouts through a customized vLLM backend (Kwon et al., 2023) and integration with the VeRL training library (Sheng et al., 2025).
- **Results:** On multi-turn tasks requiring contingent planning, EARL outperforms vocabulary-constrained baselines (*e.g.*, by 26.3% on Countdown) and, in the partially observed sorting problem, achieves perfect Sort-4 accuracy while discovering an efficient algorithm.

## 2    RELATED WORK

Recent advances demonstrate LLMs as powerful reasoners in natural language (Yao et al., 2023; Shinn et al., 2023; Schick et al., 2023). Many works have extended their role to agents interacting with external environments, and consider tasks such as tool utilization (Parisi et al., 2022; Lu et al., 2025; Qin et al., 2025), multi-modality interpretability (Zhao et al., 2024; Surís et al., 2023; Wang et al., 2024), math reasoning (Karpas et al., 2022; He-Yueya et al., 2023; Zhang et al., 2023), program-guided reasoning (Gou et al., 2024b; Gao et al., 2023; Chen et al., 2023; Liang et al.,

2023), real-time knowledge integration (Wang et al., 2025b; Gou et al., 2024a; Gu et al., 2024), and domain-specific scenarios (Bran et al., 2024; Jin et al., 2024; Theuma & Shareghi, 2024). However, most existing approaches require the model to express task-specific actions as predefined text patterns, which are then parsed and routed to external environments. This design relies heavily on the model's instruction-following ability (Hao et al., 2023), making performance highly sensitive to prompt variations (Mannekote et al., 2025) and dependent on pre-trained knowledge for action execution (Hao et al., 2023). Moreover, many methods require human-crafted demonstrations of tool usage (Chen et al., 2025a; Liu et al., 2023), further limiting scalability. In contrast, EARL endows agents with new capabilities by introducing environment-specific actions, explored and learned through counterfactual policy optimization without human demonstration.

While prior works have not explored expanding the action space of LLMs, expanding the *token space* is common in multimodal LLMs, which typically requires large-scale training on multimodal demonstrations, sometimes combined with online RL (Szot et al., 2025). Another related direction introduces action *adaptors*, which constrain the model to a set of learned actions tailored for a specific environment (Chuang et al., 2024; Wang et al., 2025c). This can be viewed as a simplified version of Figure 1b, involving only one external environment and no language environment. Beyond LLMs, growing action spaces have been studied to accelerate exploration (Farquhar et al., 2020) or to extend the set of available actions in a single environment (Jain et al., 2020; 2022; Chandak et al., 2020). Continual RL research (Khetarpal et al., 2022; Jin et al., 2023; Zhang & Lu, 2023) has also demonstrated the effectiveness of action learning in non-stationary settings (Chandak et al., 2019; Queeney et al., 2024). In contrast, our work considers the more general and challenging case where LLMs must reason in language while sequentially interacting with multiple external environments. To this end, we expand the action space of LLMs in Section 3 and propose a reinforcement learning algorithm for efficient exploration of external interactions.

## 3 FORMULATION

**Problem setting.** We consider Large Language Models (LLMs) that interact sequentially with one or more *external environments*, in addition to the default *language environment*. At each step, a model (agent) acts by selecting either a token from the language environment or an action in an external one. We formalize this as a partially observed Markov decision process (POMDP) with a global state $s_t = (h_t, e_t, z_t)$, where $h_t$ is a history of language tokens from a vocabulary $\mathcal{V}$, $e_t \in \{0, 1, \ldots, K\}$ denotes the agent's active environment ($e_t = 0$ for language), and $z_t$ is the latent state of the external environments. The history $h_t$ is fully observed by the agent, comprising a record of tokens selected in the language environment and token-based descriptions of observations from the external environments. Unlike $h_t$ and $e_t$, $z_t$ is only *partially observed* through interactions.

The agent is represented by a policy $\pi_\theta(a_t \mid h_t, e_t)$, with parameters $\theta$, that samples an action $a_t$ depending on the observable state and the active environment. Each external environment $i \neq 0$ exposes a step procedure, with a set of permissible set of actions $\mathcal{E}_i$,

$$(o_t, z_{t+1}, \textit{exit}) \quad \leftarrow \quad \text{step}_i(h_t, z_t, a_t),$$

which executes $a_t \in \mathcal{E}_i$, produces an observation described by language tokens $o_t \in \mathcal{V}^{(\cdot)}$, updates the latent state to $z_{t+1}$ and an *exit* flag. After acting, the agent updates its history by appending the new observation, $h_{t+1} \leftarrow h_t \oplus o_t$, and is routed back to the language environment if *exit* is true. At each step, the active environment produces a reward $r_t = R(e_t, h_t, a_t)$, and the agent's objective is to maximize the cumulative reward $\bar{r} = \sum_t r_t$. In some settings, the agent sets its own reward, defined in the language environment, for example, when the environment functions as a tool, *e.g.*, a calculator. In others, the language environment only facilitates reasoning capabilities, but the task and the reward are defined entirely by an external environment, for example, when sorting a list of unknown numbers by comparing and swapping symbols (see Figure 2). We describe two methods for the policy $\pi_\theta$ to issue actions to interact with environments below.

**Language-only interaction.** In existing approaches to tool use and external interaction (Feng et al., 2025; Singh et al., 2025), agents never truly leave the language environment: they always select actions $a_t \in \mathcal{V}$ from their own vocabulary, and extend the observation history as $h_{t+1} = h_t \oplus a_t$. Interacting with an external environment $i \neq 0$ requires *translating* $h_{t+1}$ to actions in $\mathcal{E}_i$. Typically, this is realized by detecting predefined patterns, such as `<calculator>...</calculator>` or structured JSON fields. When a pattern that indicates interaction with environment $i$ appears in

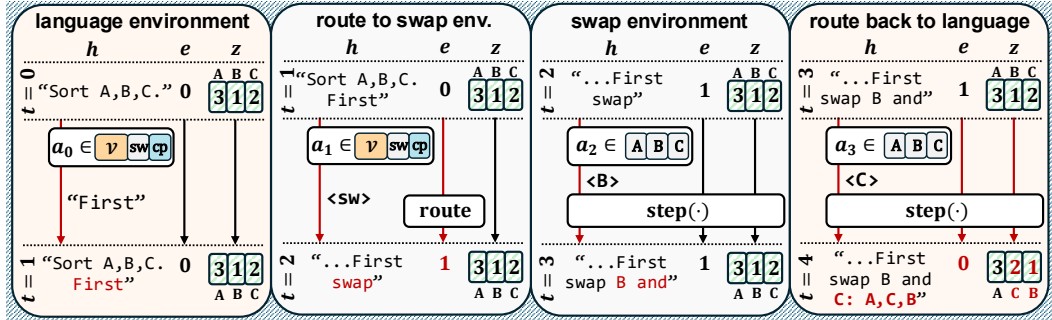

Figure 2: An example rollout with ExpA. Here, `<sw>` and `<cp>` route to the swap and compare environments, respectively. Inside them, agents can choose `<A>`, ``, `<C>` as operands. After two operands are chosen, the step procedure updates the latent state $z$ when necessary, routes back to the language environment, and returns the swap or comparison result as a plain-text observation.

$h_{t+1}$, its contents are *parsed* into a sequence of actions from $\mathcal{E}_i$ that get executed by $\text{step}_i$. A drawback is that no intermediate feedback can affect the choice of actions within the pattern (*e.g.*, `<calculator>` block), which may hamper performance and credit assignment.

**Interaction with expanded action space (ExpA).** In this work, we let agents interact directly with environments by expanding their action spaces beyond the vocabulary $\mathcal{V}$. For each environment $i \neq 0$, we add a transition action $g_i$ for entering it and an environment-specific action set $\mathcal{E}_i$ that interfaces with $\text{step}_i$. With $\mathcal{E} = \bigcup_{i=1}^{K} g_i \cup \mathcal{E}_i$ the set of added actions and $\mathcal{A} = \mathcal{V} \cup \mathcal{E}$ the full action set, the agent's policy $\pi_\theta$ is a distribution over $\mathcal{A}$, conditioned on history $h_t$ and active environment $e_t$. Algorithm 1 describes interactions under this paradigm. Every rollout begins in the language environment ($e_0 = 0$), where the policy $\pi_\theta$ may either select a vocabulary token $a_t \in \mathcal{V}$ to extend the history, or a transition action $g_i$ that *routes* control to environment $i$ while appending a description of $g_i$ to $h_t$, denoted as $\text{desc}(g_i)$. Once inside environment $i$, the policy chooses actions from $\mathcal{E}_i$, triggering the corresponding $\text{step}_i$, which outputs an observation described in $\mathcal{V}$, updates the latent state, and returns an additional *exit* flag. If $exit = \text{true}$, the environment resets to $e_{t+1} = 0$, routing control back to the language environment. We illustrate with an example in Figure 2.

**Algorithm 1** Rollout with ExpA

1: **Input:** policy $\pi_\theta$, horizon $T$, initial $s_0 = (h_0, e_0{=}0, z_0)$
2: **for** $t = 0, 1, \ldots, T - 1$ **do**
3:     **if** $e_t = 0$ **then**   ▷ language environment
4:         Sample $a_t \sim \pi_\theta(\cdot \mid h_t, e_t{=}0)$
5:         **if** $a_t \in \mathcal{V}$ **then**
6:             $h_{t+1} \leftarrow h_t \oplus a_t$;
7:         **else if** $a_t = g_i$ **then**   ▷ route to env $i$
8:             $h_{t+1} \leftarrow h_t \oplus \text{desc}(g_i)$;
9:             $e_{t+1} \leftarrow i$;
10:         **end if**
11:     **else**         ▷ external environments
12:         Sample $a_t \sim \pi_\theta(\cdot \mid h_t, e_t \neq 0)$
13:         $(o_t, z_{t+1}, exit) \leftarrow \text{step}_{e_t}(h_t, z_t, a_t)$
14:         $h_{t+1} \leftarrow h_t \oplus \text{desc}(a_t) \oplus o_t$
15:         **if** $exit{=}$true **then**
16:             $e_{t+1} \leftarrow 0$   ▷ route back to lang
17:         **end if**
18:     **end if**
19:     $r_t \leftarrow R(e_t, h_t, a_t)$
20: **end for**
21: **return** trajectory $\tau = \{(h_t, a_t, r_t)\}_{t=0}^{T-1}$

## 4   ExpA Reinforcement Learning (EARL)

An expanded action space equips LLMs with explicit means of interacting beyond language. As deciding when to route into an environment and how to act within it are inherently sequential and reward-driven tasks, we select RL as the training paradigm. We first parameterize a policy $\pi_\theta$ over the expanded action space, with careful initialization to adapt to new actions (Section 4.1). We then introduce Counterfactual Policy Optimization (CPO), which optimizes the following objective:

$$\mathcal{J}_{\text{CPO}}(\theta) = \mathbb{E}_{s_0, \, \mathcal{T}(s_0)=\{(\tau_i, \tau_i')\}_{i=1}^{m}} \left[ \frac{1}{m} \sum_{i=1}^{m} U_i(\mathcal{T}(s_0); \theta) \right], \tag{1}$$

where $s_0$ is an initial state sampled from the training distribution, $\mathcal{T}(s_0)$ comprises $m$ rollout pairs, and $U_i(\cdot; \theta)$ denotes the update function for the $i$-th pair conditioned on all rollouts. Each $\tau_i$ is a *factual* rollout obtained by inputting $s_0$ into Algorithm 1, while each $\tau_i'$ is a *counterfactual* rollout

obtained by forcing a routing action at a plausible intermediate step in $\tau_i$. We describe the construction of counterfactual rollouts in Section 4.2 and the design of the update function in Section 4.3.

## 4.1 A POLICY OVER THE EXPANDED ACTION SPACE

A central challenge in operating with expanded action spaces is how to represent and generalize to the newly introduced actions. Prior work points to two guiding principles: First, the policy should condition on the set of available actions (Jain et al., 2020; 2022). Second, prior knowledge about actions can be leveraged to improve generalization, through learned action embeddings (Jain et al., 2022) or by incorporating known structure in training (Farquhar et al., 2020). We adopt both.

**Policy parameterization.**    To condition the policy on all available actions, we extend the standard LLM classification head. In the language-only setting, the head produces $|\mathcal{V}|$ logits over the vocabulary. With ExpA, this head is expanded to output $|\mathcal{V} \cup \mathcal{E}|$ logits. We denote by $\theta$ the parameters of the LLM together with the expanded head. At step $t$, the encoded feature of $h_t$ is projected to logits, and a softmax is applied over the subset of actions available in environment $e_t$, yielding $\pi_\theta(\cdot \mid h_t, e_t)$.

**Policy initialization.**    Each action $a \in \mathcal{E}$ has a natural language description $\mathrm{desc}(a)$, such as the environment name (*e.g.*, "calculator") or the semantic label of a step procedure (*e.g.*, "compare"). To exploit this prior knowledge about action similarities, we initialize the weights of new actions so that selecting an action has approximately the same likelihood as producing its description:

$$\pi_\theta(a \mid h_t, e_t) \approx \pi_\theta(\mathrm{desc}(a) \mid h_t, e_t),$$

where $e_t$ is the active environment at $t$. In particular, when the description is a single token, this condition can be satisfied directly (and exactly) by initializing the new action weight with the pretrained weight of the token $\mathrm{desc}(a)$. This aligns expanded actions with their linguistic counterparts from the start, providing a strong prior that accelerates learning.

## 4.2 COUNTERFACTUAL ROLLOUTS TO ENCOURAGE INVOKING NEW ACTIONS

Even with careful initialization and prompting about environments, the policy may fail to reliably invoke routing actions when needed. For instance, a pretrained model has no prior experience of invoking a calculator and thus may not assign high probability to its routing action (*e.g.*, $g_{\mathrm{calc}} =$ `<calculate>`), even when complex arithmetic is required. We address this with *counterfactual rollouts*, which evaluate *what would have happened* had the policy taken a routing action at a plausible intermediate step, thereby encouraging exploration of rarely invoked but critical decisions.

Given a factual rollout $\tau = \{(h_t, a_t, r_t)\}_{t=0}^{T-1}$, we construct a counterfactual rollout $\tau'$ as follows: 1) Select a routing action $g_i \in \mathcal{E}$ to be encouraged (*e.g.*, $g_{\mathrm{calc}}$ for arithmetic tasks); 2) Sample a time step $t' \in \{t \mid e_t = 0\}$ with weight proportional to $\pi_\theta(\mathrm{desc}(g_i) \mid h_t, e_t = 0)$; 3) Initialize $\tau'_t \leftarrow \tau_t$ using the factual rollout for $t = 1, ..., t'$; 4) Intervene with $\mathrm{do}(a_{t'} = g_i)$ at $t'$ and apply the transition in Algorithm 1; 5) Continue rollout for $t = t' + 1, \ldots, T - 1$ with Algorithm 1 to obtain $\tau'$.

This relies *only* on the pretrained next-token distribution (step 2). For example, if $\mathrm{desc}(g_{\mathrm{calc}}) =$ "calculate", the insertion "To solve it, first calculate" is more probable under the language model than "To solve calculate, ..." and is weighed more heavily when forcing a routing action. Hence the method is fully compatible with zero-RL training (DeepSeek-AI et al., 2025; Zeng et al., 2025).

## 4.3 UPDATE FUNCTION

Finally, we define the update function for each rollout pair $(\tau_i, \tau'_i)$, $i \in \{1, \ldots, m\}$, as

$$U_i(\mathcal{T}(s_0); \theta) = \begin{cases} f(\tau'_i, \overline{r}'_i - \overline{r}_i; \theta), & \text{if } \overline{r}_j \leq 0 \ \forall j \in \{1, \ldots, m\}, \\ f(\tau_i, \frac{\overline{r}_i - \mu}{\sigma}; \theta), & \text{otherwise,} \end{cases}$$

where $\overline{r}_i$ and $\overline{r}'_i$ denote the cumulative rewards of $\tau_i$ and $\tau'_i$, respectively, and $\mu, \sigma$ are the mean and standard deviation of rewards across the factual rollouts. Here $f(\tau, a; \theta)$ denotes the standard update rule (Shao et al., 2024), which takes as input a rollout trajectory $\tau$ and its associated advantage scalar, and applies PPO-style clipping and KL regularization (details in Appendix D). The design is motivated by balancing exploration and exploitation: when the current rollout fails to achieve positive reward, the first counterfactual branch encourages exploration of missing interactions; otherwise, the update reduces to group-relative advantage, exploiting successful strategies.

Table 1: Statistics of the Calc-Bench datasets. Language portions refer to the portion of questions where operations or numbers are written in natural language.

| Task | Max number ($10^x$) | | #Operands | | Lang. portion | | #Instances | |
|------|-------|------|-------|------|-------|------|--------|--------|
| | Train | Test | Train | Test | Train | Test | Train | Test |
| Arithmetic | 5 | 5 | 5 | 7 | 10% | 70% | 1,000 | 2,000 |
| Countdown | 4 | 4 | 4 | 4 | NA | NA | 20,000 | 2,000 |
| GSM8K* | 6 | 6 | NA | NA | NA | NA | 5,317 | 579 |
| Count | 20 | 20 | NA | NA | 90% | 90% | 1,000 | 2,000 |

**Example Questions**

**Arithmetic**   What is subtraction of 69 divided by 6 plus 7049 plus 13 plus 50 plus 643 from 9936?

**Countdown**   Using the numbers [2900, 1205, 5911, 4], create an equation that equals -4212.

**GSM8K***   Henry made two stops in a 691206-mile trip...How many miles did he go in the first stop?

**Count**   How many times does the digit 0 appear in the number fifty-eight billion, thirty million...?

**Sorting**   Sort the following items in ascending order: A, B, C, D, E.

**Environments**

**Calculator**
$+-\times\div= ()1234567890.$

**Compare**
$A, B, C, D, E$

**Swap**
$A, B, C, D, E$

Figure 3: Example questions and the environment-specific actions in Calc-Bench and Sorting.

## 5 EXPERIMENT

### 5.1 EXPERIMENTAL SETUP

We evaluate our method in settings that require multi-turn interactions with external environments. Prior works evaluate mainly on math problems (Feng et al., 2025) or API calls (Qin et al., 2024), where the solution path follows a *fixed* derivation. We additionally stress the more challenging *contingent planning* problem, where the agent must adapt its actions on the fly based on intermediate observations from interactions. To this end, we design two complementary tasks below:

- **Calc-Bench.** The agent has access to a stateless *calculator* environment (Figure 3) that provides arithmetic knowledge to support language reasoning in tasks where a reward is given if the final solution exactly matches (EM) the target. The benchmark comprises four types of tasks, described by example in Figure 3 (statistics in Table 1): (1) *Arithmetic* tests calculator use out-of-distribution by varying the number of operands and the amount of natural language used in problem instances. This requires a robust mapping between language and numbers. (2) *Countdown* stresses contingent planning: each problem admits up to 7,680 unique combinations, forcing the agent to reason efficiently by adjusting its strategy based on intermediate outcomes (*e.g.*, aggressively adjusting strategy when far from the target). (3) *GSM8K** enhances the widely used GSM8K Cobbe et al. (2021) by scaling up the numbers while preserving problem semantics, increasing difficulty and requiring accurate understanding of the text and its translation into computational steps. (4) *Count* requires the agent to preserve its basic numerical understanding while learning tasks (1)-(3).

- **Sorting.** The agent must arrange a set of *hidden* numbers in ascending or descending order by interacting with *compare* and *swap* environments (Figure 3); for example, "compare $A, B$" reveals their relative order, while "swap $A, B$" updates the *hidden state* $z_t$ by exchanging their positions (Figure 2). The reward depends on the final *hidden state* $z_T$ being correctly sorted, with penalties for excessive numbers of comparisons and swaps. This setting is particularly challenging, as it forms a POMDP that requires contingent planning based on intermediate comparison results, while also demanding *precise* and *efficient* reasoning over first-order logic relations. Moreover, the agent must uncover and modify the hidden state through environment interactions rather than simply outputting a textual answer, making this a realistic testbed for interactive decision-making situations such as embodied AI. Training data consists of sorting problems of different sizes (Sort-2 to Sort-5) and testing evaluates on Sort-4 and Sort-5. Other details are in the Appendix E.

**Baselines.** We compare against baselines that reflect distinct learning paradigms: (1) *SFT+GRPO*: the model is first fine-tuned on labeled interaction data and then optimized with Group Relative Policy Optimization (GRPO) (Shao et al., 2024), following the setup in (Feng et al., 2025). (2) *Prompt+GRPO* (DeepSeek-AI et al., 2025): environment interaction patterns are provided in the prompt, and the model is further optimized with GRPO, as in (Singh et al., 2025). (3) *Prompt+CPO*: similar to Prompt+GRPO, but trained with our Counterfactual Policy Optimization (CPO) instead of

Table 2: EM results (exact match) on Calc-Bench. We train each model jointly on all Calc-Bench tasks to assess the benefits of shared representation learning.

| Method | Calc-Bench | | | | |
|---|---|---|---|---|---|
| | Arithmetic | Countdown | Count | GSM8K* | Overall |
| **GPT-4o** | 41.30 | 18.85 | 66.85 | 31.95 | 39.74 |
| **Qwen-2.5-3B-Instruct** | 15.80 | 2.80 | 66.50 | 20.55 | 26.41 |
| SFT+GRPO | **70.75** | 48.50 | 93.85 | 30.57 | 60.92 |
| Prompt+GRPO | 64.70 | 49.15 | **94.75** | 30.39 | 59.75 |
| Prompt+CPO | 61.50 | 38.30 | 91.35 | 46.80 | 59.49 |
| ExpA+CPO (**EARL**) | 69.20 | **75.15** | 93.70 | **48.53** | **71.65** |
| **Qwen-2.5-7B-Instruct** | 22.60 | 11.75 | 74.05 | 24.01 | 33.10 |
| SFT+GRPO | 56.00 | 66.70 | 93.00 | 34.20 | 62.48 |
| Prompt+GRPO | **80.30** | 60.70 | 98.60 | 33.33 | 68.23 |
| Prompt+CPO | 64.85 | 55.15 | 94.55 | 52.33 | 66.72 |
| ExpA+CPO (**EARL**) | 78.10 | **84.25** | **98.70** | **53.71** | **78.69** |

Table 3: Average occurrence of external interactions, hallucinations and planning phrases in a validation rollout on Countdown task. We define hallucination as inputting a number that is not given in the question to the calculator. We identify a list of planning phrases commonly used by models in Appendix I.1.

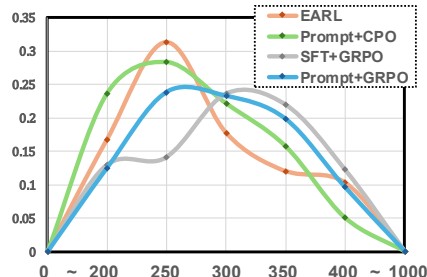

| Method | #interactions | #hallucinations | #plannings |
|---|---|---|---|
| EARL | 11.25 | 0.001 | 20.48 |
| Prompt+CPO | 4.85 | 0.7424 | 3.22 |
| Prompt+GRPO | 15.10 | 0.000 | 1.84 |
| SFT+GRPO | 3.16 | 0.6885 | 1.96 |

Figure 4: Response token length distribution for correct rollouts in GSM8K*.

GRPO, so that the only difference frosm EARL is the absence of expanded action space. (4) *Zero-shot*: evaluation of proprietary models such as GPT-4o (Hurst et al., 2024) without any fine-tuning.

**Implementation Details.** We use the open-source Qwen2.5 (Yang et al., 2024) as our backbone, including results for both base models and instruction-tuned variants with 0.5B, 3B, and 7B parameters. The maximum sequence length is set to 1,024 for Calc-Bench and 384 for Sorting. Training is performed on NVIDIA A100-80GB GPUs, with 1, 2, and 4 GPUs allocated for the 0.5B, 3B, and 7B models, respectively. For fair comparison, we follow standard hyperparameters and optimization protocols (Singh et al., 2025) (*e.g.*, KL regularization weight, PPO clipping threshold) for both baselines and our method, with full details provided in the Appendices F and G.

## 5.2 EXPERIMENTAL RESULTS ON CALC-BENCH

**Main results.** As shown in Table 2, zero-shot models perform poorly on this challenging benchmark, with GPT-4o reaching only 39.74 overall EM. Training with a calculator greatly improves performance, but baselines remain inconsistent across tasks and perform especially poorly on contingent-planning tasks like Countdown. In contrast, EARL delivers strong results across all tasks, with up to 10.46 absolute EM gain overall and as much as 17.55 EM gain on Countdown.

**Countdown analysis.** We provide a detailed analysis of the results on Countdown with the 3B model in Table 3, supplemented by case studies in the Appendix J. Several observations emerge:

- Prompt+GRPO triggers the most calculator interactions and avoids hallucinations. However, it often degenerates into inefficient brute-force trials, showing limited use of planning cues after observations (*e.g.*, "this is far from target").

- Replacing GRPO with CPO increases the use of planning keywords, likely because counterfactual interventions provide more training signals on how to react to observations. Yet this also introduces hallucinations, where the agent invents numbers not in the problem. A plausible cause is

Table 4: Ablation on Countdown: CPO vs. GRPO, training on Qwen-Instruct vs. -Base, and with (w/) vs. without (w/o) environment prompt (env.p). ExpA+CPO corresponds to EARL.

|  | Instruct | | Base | |
| --- | --- | --- | --- | --- |
|  | w/ env.p | w/o | w/ env.p | w/o |
| ExpA+CPO | 80.09 | 76.76 | 77.31 | 74.56 |
| ExpA+GRPO | 75.10 | 73.79 | 76.45 | 70.27 |
| Prompt+CPO | 67.23 | - | 63.64 | - |
| Prompt+GRPO | 58.16 | - | 51.15 | - |
| SFT+GRPO | 62.05 | - | 61.17 | - |

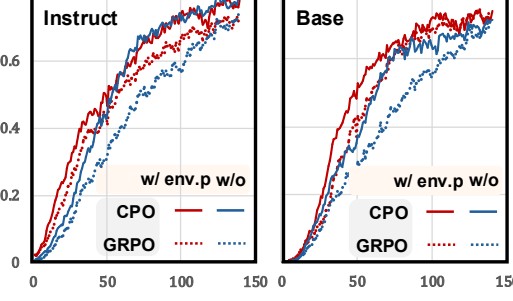

Figure 5: ExpA training reward vs. iteration.

interference between language reasoning and action learning, *e.g.*, the KL penalty helps preserve pre-trained language knowledge but may hinder the grounding of calculator actions.

- SFT+GRPO uses the fewest interactions, sometimes performing parts of a computation in language mode and then feeding incorrect results to the calculator, causing hallucinations. This suggests that SFT-learned patterns transfer poorly to diverse problem instances.

- EARL uses a moderate number of interactions but produces by far the most planning-related language, yielding much stronger results than all baselines in Table 2. This strongly validates the benefit of decoupling environment interactions from language reasoning, which removes confusion between reasoning and action learning and enables effective use of external environments.

**GSM8K$^*$ analysis.** In Figure 4, we show the distribution of response lengths among correct rollouts on GSM8K$^*$ with the 3B model. The results reveal a clear link between efficiency in reasoning (fewer tokens) and stronger performance. Notably, the two methods using CPO (EARL and Prompt+CPO) outperform those with GRPO (SFT+GRPO and Prompt+GRPO), underscoring the importance of encouraging diverse environment interactions.

**Ablation Study.** We perform ablation on the challenging Countdown task from 3 perspectives:

- *CPO vs. GRPO*: As shown in Table 4, CPO consistently outperforms GRPO across all settings, even for baselines (Prompt+CPO). It also converges faster, as seen by comparing each solid line (CPO) with the dotted line of the same color (GRPO) in Figure 5, highlighting the role of counterfactual rollouts in promoting exploration of new environments and their actions.

- *Instruct vs. Base*: With ExpA, even base models achieve competitive performance, whereas baselines algorithms such as Prompt+GRPO degrade sharply without instruction tuning. This suggests strong potential for using EARL in Zero-RL training of agents for interactive problem solving.

- *Prompted vs. unprompted environments*: Prompting the agent on how to interact is essential for prompt+RL baselines. With ExpA, however, models succeed without such prompts by leveraging weight initialization (Section 4.1, ExpA+GRPO w/o) and counterfactual rollouts (Section 4.2, ExpA+CPO w/o), indicating scalability to settings with large number of environments.

**Additional results.** We provide results on the 0.5B model in the Appendix H.1 to demonstrate that it can interact with env as well. We also provide the change of validation performance throughout training on Calc-Bench, as well as case studies that highlight our advantages over previous methods.

## 5.3 EXPERIMENTAL RESULTS ON SORTING

**Main results.** Figure 6 reports sorting accuracy for Prompt+GRPO, Prompt+CPO, and EARL on Sort-4 and Sort-5 tasks, stratified by the minimum number of swaps required. On Sort-4, EARL achieves perfect accuracy across all levels, whereas Prompt+CPO degrade as the required number of swaps increases. The gap widens on the more challenging Sort-5 problems, where EARL outperforms the best baseline (Prompt+GRPO) by up to 21% in a stratum and more than 10% overall.

**Efficiency.** We evaluate the efficiency of EARL's learned sorting strategy by measuring the average number of comparisons and swaps needed to sort random numbers. Under greedy decoding, EARL follows a deterministic decision tree, which we visualize in the Appendix H.2. By pruning a few redundant comparisons, we obtain a simplified variant, EARL$^*$, corresponding to the algorithm

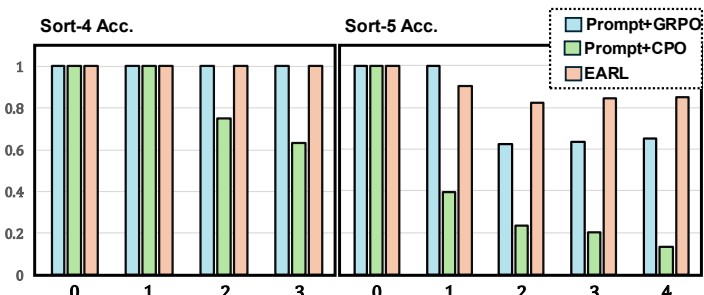

Figure 6: Sort accuracy stratified by the number of required swaps.

Table 5: Average number of swaps (SW) and comparisons (CP) to sort 4 random numbers for different algorithms.

| Method | #SW | #CP |
|---|---|---|
| Prompt+GRPO | 2.076 | 6.101 |
| EARL | 1.917 | 5.708 |
| EARL$^*$ | 1.917 | 4.833 |
| GCC | 2.333 | 5.000 |
| Insert-sort | 3.000 | 4.917 |
| Optimal | 1.917 | 4.667 |

in Algorithm 2. This algorithm uses the element $A$ as a pivot to compare against the other elements, performs additional checks only when necessary, and finally applies MIN_SWAP($\mathcal{R}$) to sort with minimal swaps given the accumulated comparison results $\mathcal{R}$. We compare EARL and EARL$^*$ against Prompt+GRPO, classical sorting algorithms and the theoretical optimum in Table 5. Both variants exactly match the optimal number of swaps and closely approach the optimal number of comparisons, outperforming insertion sort and even GCC's built-in routine.[1]

**Sorting with RL.** Our study connects to recent work on discovering faster sorting algorithms with RL, most notably AlphaDev (Mankowitz et al., 2023). A key distinction is that we leverage the LLM's natural language vocabulary to represent context and chain reasoning steps, rather than relying on dedicated symbolic states or low-level assembly instructions. This means that our agent is more general-purpose, capable of reusing pre-trained language knowledge. Consequently, EARL achieves 100% accuracy on Sort-4 after only $\sim$**70** training steps, compared to the million-step training required by AlphaDev, underscoring the value of transferring language reasoning into interactive environments. While performance on Sort-5 is not yet perfect, our goal is to demonstrate how ExpA improves reasoning with external environments, leaving dedicated algorithm discovery and more challenging settings (*e.g.*, VARSORT) as promising future work.

---

**Algorithm 2** EARL$^*$ Sort-4

1: **Input:** four numbers $A, B, C, D$
2: $\mathcal{R} \leftarrow \emptyset$
3: $\mathcal{R} \leftarrow \mathcal{R} \cup \{\text{Compare}(A, B)\}$
4: $\mathcal{R} \leftarrow \mathcal{R} \cup \{\text{Compare}(A, C)\}$
5: $\mathcal{R} \leftarrow \mathcal{R} \cup \{\text{Compare}(A, D)\}$
6: **if** not $(C < A < B \ \lor \ B < A < C)$ **then**
7: $\quad \mathcal{R} \leftarrow \mathcal{R} \cup \{\text{Compare}(B, C)\}$
8: **end if**
9: **if** not $(D < A < B \ \lor \ B < A < D)$ **then**
10: $\quad \mathcal{R} \leftarrow \mathcal{R} \cup \{\text{Compare}(B, D)\}$
11: **end if**
12: **if** not $(D < A < C \ \lor \ C < A < D)$ **then**
13: $\quad \mathcal{R} \leftarrow \mathcal{R} \cup \{\text{Compare}(C, D)\}$
14: **end if**
15: MIN_SWAP($\mathcal{R}$)

---

# 6 CONCLUSION

We have introduced a new paradigm for enabling Large Language Models (LLMs) to reason with and beyond language when interfacing with external environments. Our proposed framework, **ExpA**, introduces routing and environment-specific actions to decouple reasoning from interaction. This removes the reliance on external parsers in the current language-only paradigm to detect special interaction language syntax, and hence enables true end-to-end training. To optimize policies for interactive problem solving, we proposed **EARL**, a reinforcement learning method based on counterfactual rollouts that encourages exploration of new and rarely used, but critical environment interactions. Empirically, EARL outperforms strong baselines on multi-turn tasks that benefit from or require environment interaction, with particular gains in challenging settings that demand contingent planning. It also shows consistent improvements in multi-task and continual learning scenarios, and notably discovers an efficient algorithm for sorting with four elements. This work establishes a scalable and principled framework for equipping LLMs with explicit capabilities to interact with external environments, opening future directions in mathematical reasoning, embodied AI, continual learning, and large-scale zero-RL training with tools.

---

[1]We note that GCC is optimized for processor efficiency (*e.g.*, branchless `cmov`) rather than minimizing raw operations, whereas "Optimal" denotes the theoretical minimum over swaps and comparisons combined.

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

CONTENTS OF APPENDIX

# A  LIMITATIONS AND FUTURE DISCUSSIONS

The primary objective of this study is to develop a practical method for extending the action space of LLMs, thereby enhancing their reasoning capabilities beyond the intrinsic language knowledge spaces. Our proposed approach, EARL, proves effective in this regard. However, we identify three key limitations. First, due to computational constraints, our empirical evaluation is limited to the Qwen-2.5 model family (up to 7B parameters), and its scaling properties remain to be discovered. Second, we did not investigate optimal action space initialization techniques for novel actions within the continuous learning paradigm. Third, the framework's performance in complex external environments involving multimodality and diverse action spaces is underexplored. Addressing these limitations to develop a more robust and generalizable solution will be the focus of our future research endeavors.

# B  USAGE OF AI ASSISTANT

We use AI assistants or tools such as ChatGPT and Grammarly to correct grammar errors and polish the language.

# C  POLICY ON EXPANDED ACTION SPACE

Our parameterization of the $\pi_\theta$ uses a linear-softmax layer applied to an encoding $g(h_t) \in \mathbb{R}^d$ of the history (context) $h_t$ by the LLM. The layer uses a weight matrix $W = [w_1, ..., w_N]^\top$, where $N = |\mathcal{E}|$ is the total number of actions across all environments, including the language one, and $w_a \in \mathbb{R}^d$ are the parameters used to compute the logit of action $a$ for $a \in [N]$. The softmax is restricted to actions that are available in the active environment $e_t$. We define, for environments $e \in \{0, 1, ..., K\}$,

$$\sigma_e(z)_a = \frac{e^{z_a}}{\sum_{a' \in \tilde{\mathcal{E}}_e} e^{z_{a'}}} \mathbb{1}[a \in \tilde{\mathcal{E}}_e] .$$

Here, $\tilde{\mathcal{E}}_0 = \{g_e\}_{e \in [K]} \cup \mathcal{E}_0$ and $\tilde{\mathcal{E}}_e = \mathcal{E}_e$ for $e = 1, ..., K$. With this,

$$\pi_\theta(\cdot \mid h_t, e_t) = \sigma_{e_t}(W g(h_t)) .$$

# D  UPDATE RULE

The update rule is given by

$$f(\tau, a; \theta) = \sum_{t=0}^{T-1} \Big[ \min\big(r_t(\theta)\, a,\ \mathrm{clip}(r_t(\theta), 1 - \epsilon, 1 + \epsilon)\, a\big)\ -\ \beta\, \mathrm{KL}\big(\pi_\theta(\cdot \mid h_t) \,\|\, \pi_{\mathrm{ref}}(\cdot \mid h_t)\big) \Big],$$

(2)

where

- $\tau = \{(h_t, a_t, r_t)\}_{t=0}^{T-1}$ is a rollout trajectory,
- $h_t$ denotes the token history (state) at step $t$,
- $r_t(\theta) = \frac{\pi_\theta(a_t|h_t)}{\pi_{\mathrm{old}}(a_t|h_t)}$ is the importance sampling ratio between the new and old policies,
- $\epsilon$ is the PPO clipping threshold,
- $\beta$ is the KL regularization coefficient,
- $\pi_\theta(\cdot \mid h_t)$ and $\pi_{\mathrm{ref}}(\cdot \mid h_t)$ denote the current and reference policies, respectively. We use the pre-trained LLM as the reference model.

Note that we slightly abuse notation in the main paper by saying $\tau$ is generated with $\pi_\theta$. In practice, rollouts are generated by the reference policy $\pi_{\mathrm{old}}$, which is held fixed during data collection. The objective in Equation (2) then compares the likelihood of these sampled actions under the current policy $\pi_\theta$ versus the reference policy $\pi_{\mathrm{old}}$, with the ratio $r_t(\theta)$ providing the necessary importance weighting. This distinction ensures stable on-policy learning: trajectories are collected with $\pi_{\mathrm{old}}$, while updates adjust $\pi_\theta$ to maximize advantage without diverging too far from $\pi_{\mathrm{old}}$.

In training, we do not perform update on positions in each rollout corresponding to observations returned by external environments. For EARL, we apply the KL loss with token probabilities computed over the original vocabulary space $\mathcal{V}$.

# E    DATASET DETAILS

**Calc-Bench Dataset Details.**    The *Calc-Bench* benchmark consists of four sub-datasets targeting different types of mathematical reasoning: *Arithmetic*, *Countdown*, *Count*, and a rewritten subset of GSM8K, denoted *GSM8K**. Each dataset is generated via task-specific scripts, with explicit control over complexity, number format, and structure.

- *Arithmetic*: This dataset contains randomly generated arithmetic expressions involving up to 5-digit integers and between 2 and 6 operations. The training set includes 1,000 samples with a maximum of 4 operations and 10% of examples paraphrased into natural language. The test set contains 2,000 samples with up to 6 operations and 70% paraphrased into natural language.

- *Countdown*: Each instance is generated by sampling a valid arithmetic expression, extracting all numerical values, shuffling them, and using the original result as the target. The final dataset includes 20,000 training and 2,000 test examples. Expressions contain up to 4-digit numbers and allow up to 3 operations. Each number must appear exactly once in the constructed expression.

- *GSM8K**: This subset is derived from a manually rewritten version of GSM8K, filtered to include only examples explicitly marked as rewritten. Each sample includes a paraphrased question and corresponding reasoning-based answer. This version preserves the complexity of GSM8K while reducing lexical overlap with the original dataset.

- *Count*: This dataset consists of symbol sequences with a maximum length of 20. Each example is labeled with a count-based target (e.g., the number of specific items). The training set includes 1,000 examples and the test set includes 2,000.

**Sorting Dataset Details.**    We adopt a curriculum-based training strategy for sorting tasks. Specifically, we first use an *ordering* dataset to pre-train the model on simpler relational tasks, followed by a *sorting* dataset that introduces the full sorting items. Each dataset is constructed with its own generation procedure and input length distribution, as detailed below.

- *Ordering*: A total of 20,000 training examples are generated, with 95% assigned to the *order* task and 5% to *compare*. Each input sequence contains 2 to 5 items, with a length distribution of 30% for 2 items, 30% for 3 items, 20% for 4 items, and 20% for 5 items. The corresponding test set contains 2,000 *order* examples, with the same item length distribution.

- *Sorting*: The training set contains 80,000 examples, with input lengths ranging from 2 to 5 elements, denoted as Sort-2 through Sort-5. The distribution is 10% Sort-2, 20% Sort-3, 30% Sort-4, and 40% Sort-5. The test set includes 2,000 examples, equally split between Sort-4 and Sort-5 cases.

# F    BASELINE DETAILS

In our experiments, we employ open-source LLM Qwen2.5 (Yang et al., 2024) as the backbone, which has been designed to address a wide range of applications, including coding and mathematics. We select both base and instruction-tuned variants with model sizes of 0.5B, 3B, and 7B parameters for our experiments.

Here we provide details of each baseline methods as follows:

- **GPT-4o** (Hurst et al., 2024) is a frontier large language model developed by OpenAI, demonstrating advanced capabilities in reasoning with various contexts. We use standard prompt instruction as the key to generating a response.

- **Prompt-based RL** is a cost-effective and effective method for LLMs, specifically designed to train them by optimizing their responses to given prompts. We employ the cutting-edge approach Group Relative Policy Optimization (GRPO) (Shao et al., 2024) for our **SFT+GRPO** and **Prompt-**

**GRPO** (DeepSeek-AI et al., 2025) baselines. We replace GRPO with our proposed CPO as **Prompt-CPO** baseline.

**SFT Data Curation.** We follow Tang et al. (2024), leveraging frontier LLM GPT-4o to generate solutions on the Countdown task. In order to obtain high-quality SFT data, we only select solutions that GPT-4o can correctly answer with no more than 1K tokens.

# G    IMPLEMENTATION DETAILS

Our implementation of EARL efficiently supports ExpA rollouts through a customized vLLM backend (Kwon et al., 2023) and integration with the VeRL training library (Sheng et al., 2025). We follow the configurations outlined in VeRL. To ensure the reproducibility of our findings, detailed implementation instructions are provided below.

## G.1    CALC-BENCH IMPLEMENTATION DETAILS

```
Prompt+GRPO Training Config <Qwen-2.5-3B-Instruct>
apptainer exec --nv envs/verl.sif bash -c "
    ray start --head --port=6383 &&
    set -x &&
    python3 -m verl.trainer.main_earl
    actor_rollout_ref.earl.model.freeze_base_model=False
    actor_rollout_ref.earl.model.init_from_base=True
    actor_rollout_ref.earl.training.tools=['calculator']
    algorithm.adv_estimator=grpo
    tool_config_path=tool_configs/combined_calculator_baseline.yaml
    data.train_files=./data/calc_bench/combined_baseline/train.parquet
    data.val_files=./data/calc_bench/combined_baseline/test.parquet
    data.train_batch_size=256
    data.max_prompt_length=384
    data.max_response_length=1024
    data.filter_overlong_prompts=True
    data.truncation='error'
    actor_rollout_ref.model.path=Qwen/Qwen2.5-3B-Instruct
    actor_rollout_ref.actor.optim.lr=1e-6
    actor_rollout_ref.model.use_remove_padding=True
    actor_rollout_ref.actor.ppo_mini_batch_size=32
    actor_rollout_ref.actor.ppo_micro_batch_size_per_gpu=8
    actor_rollout_ref.actor.use_kl_loss=True
    actor_rollout_ref.actor.kl_loss_coef=0.001
    actor_rollout_ref.actor.kl_loss_type=low_var_kl
    actor_rollout_ref.actor.entropy_coeff=0
    actor_rollout_ref.model.enable_gradient_checkpointing=True
    actor_rollout_ref.actor.entropy_from_logits_with_chunking=True
    actor_rollout_ref.actor.fsdp_config.param_offload=False
    actor_rollout_ref.actor.fsdp_config.optimizer_offload=False
    actor_rollout_ref.rollout.log_prob_micro_batch_size_per_gpu=8
    actor_rollout_ref.rollout.tensor_model_parallel_size=1
    actor_rollout_ref.rollout.name=vllm
    actor_rollout_ref.rollout.gpu_memory_utilization=0.6
    actor_rollout_ref.rollout.n=8
    actor_rollout_ref.ref.log_prob_micro_batch_size_per_gpu=8
    actor_rollout_ref.ref.fsdp_config.param_offload=True
    algorithm.use_kl_in_reward=False
    trainer.critic_warmup=0
    trainer.logger=['console','tensorboard']
    trainer.project_name='earl'
    trainer.experiment_name='calc_bench/qwen2.5-3b/prompt-grpo'
    trainer.validation_data_dir=./val_result/calc_bench/3b/prompt-grpo
    trainer.n_gpus_per_node=4
    trainer.nnodes=1
    trainer.save_freq=25
```

```
    trainer.test_freq=25
    trainer.total_epochs=4
```

```
──────────── EARL Training Config <Qwen-2.5-3B-Instruct> ────────────
apptainer exec --nv envs/verl.sif bash -c "
    ray start --head --port=6383 &&
    set -x &&
    python3 -m verl.trainer.main_earl
    actor_rollout_ref.earl.model.freeze_base_model=False
    actor_rollout_ref.earl.model.init_from_base=True
    actor_rollout_ref.earl.training.tools=['calculator']
    algorithm.adv_estimator=trpo
    tool_config_path=tool_configs/combined_calculator.yaml
    data.train_files=./data/calc_bench/combined_earl/train.parquet
    data.val_files=./data/calc_bench/combined_earl/test.parquet
    data.train_batch_size=256
    data.max_prompt_length=384
    data.max_response_length=1024
    data.filter_overlong_prompts=True
    data.truncation='error'
    actor_rollout_ref.model.path=Qwen/Qwen2.5-3B-Instruct
    actor_rollout_ref.actor.optim.lr=1e-6
    actor_rollout_ref.model.use_remove_padding=True
    actor_rollout_ref.actor.ppo_mini_batch_size=32
    actor_rollout_ref.actor.ppo_micro_batch_size_per_gpu=8
    actor_rollout_ref.actor.use_kl_loss=True
    actor_rollout_ref.actor.kl_loss_coef=0.001
    actor_rollout_ref.actor.kl_loss_type=low_var_kl
    actor_rollout_ref.actor.entropy_coeff=0
    actor_rollout_ref.model.enable_gradient_checkpointing=True
    actor_rollout_ref.actor.entropy_from_logits_with_chunking=True
    actor_rollout_ref.actor.fsdp_config.param_offload=False
    actor_rollout_ref.actor.fsdp_config.optimizer_offload=False
    actor_rollout_ref.rollout.max_num_batched_tokens=7200
    actor_rollout_ref.rollout.log_prob_micro_batch_size_per_gpu=8
    actor_rollout_ref.rollout.tensor_model_parallel_size=1
    actor_rollout_ref.rollout.name=vllm
    actor_rollout_ref.rollout.gpu_memory_utilization=0.6
    actor_rollout_ref.rollout.n=8
    actor_rollout_ref.ref.log_prob_micro_batch_size_per_gpu=8
    actor_rollout_ref.ref.fsdp_config.param_offload=True
    algorithm.use_kl_in_reward=False
    trainer.critic_warmup=0
    trainer.logger=['console','tensorboard']
    trainer.project_name='earl'
    trainer.experiment_name='calc_bench/qwen2.5-3b/earl-trpo'
    trainer.validation_data_dir=./val_result/calc_bench/3b/earl
    trainer.n_gpus_per_node=4
    trainer.nnodes=1
    trainer.save_freq=25
    trainer.test_freq=25
    trainer.total_epochs=4
```

```
─────── Prompt+GRPO Training Config <Qwen-2.5-7B-Instruct> ───────
apptainer exec --nv envs/verl.sif bash -c "
    ray start --head --port=6384 &&
    set -x &&
    python3 -m verl.trainer.main_earl
    actor_rollout_ref.earl.model.freeze_base_model=False
    actor_rollout_ref.earl.model.init_from_base=True
    actor_rollout_ref.earl.training.tools=['calculator']
    algorithm.adv_estimator=grpo
```

```
        tool_config_path=tool_configs/combined_calculator_baseline.yaml
        data.train_files=./data/calc_bench/combined_baseline/train.parquet
        data.val_files=./data/calc_bench/combined_baseline/test.parquet
        data.train_batch_size=256
        data.max_prompt_length=384
        data.max_response_length=1024
        data.filter_overlong_prompts=True
        data.truncation='error'
        actor_rollout_ref.model.path=Qwen/Qwen2.5-7B-Instruct
        actor_rollout_ref.actor.optim.lr=1e-6
        actor_rollout_ref.model.use_remove_padding=True
        actor_rollout_ref.actor.ppo_mini_batch_size=32
        actor_rollout_ref.actor.ppo_micro_batch_size_per_gpu=8
        actor_rollout_ref.actor.use_kl_loss=True
        actor_rollout_ref.actor.kl_loss_coef=0.001
        actor_rollout_ref.actor.kl_loss_type=low_var_kl
        actor_rollout_ref.actor.entropy_coeff=0
        actor_rollout_ref.model.enable_gradient_checkpointing=True
        actor_rollout_ref.actor.entropy_checkpointing=True
        actor_rollout_ref.actor.entropy_from_logits_with_chunking=True
        actor_rollout_ref.actor.fsdp_config.param_offload=False
        actor_rollout_ref.actor.fsdp_config.optimizer_offload=False
        actor_rollout_ref.rollout.log_prob_micro_batch_size_per_gpu=8
        actor_rollout_ref.rollout.tensor_model_parallel_size=1
        actor_rollout_ref.rollout.name=vllm
        actor_rollout_ref.rollout.gpu_memory_utilization=0.6
        actor_rollout_ref.rollout.n=8
        actor_rollout_ref.ref.log_prob_micro_batch_size_per_gpu=32
        actor_rollout_ref.ref.fsdp_config.param_offload=True
        algorithm.use_kl_in_reward=False
        trainer.critic_warmup=0
        trainer.logger=['console','tensorboard']
        trainer.project_name='earl'
        trainer.experiment_name='calc_bench/qwen2.5-7b/prompt-grpo'
        trainer.validation_data_dir=./val_result/calc_bench/7b/prompt-grpo
        trainer.n_gpus_per_node=4
        trainer.nnodes=1
        trainer.save_freq=100
        trainer.test_freq=10
        trainer.total_epochs=4
```

```
                    EARL Training Config <Qwen-2.5-7B-Instruct>
apptainer exec --nv envs/verl.sif bash -c "
    ray start --head --port=6384 &&
    set -x &&
    python3 -m verl.trainer.main_earl
    actor_rollout_ref.earl.model.freeze_base_model=False
    actor_rollout_ref.earl.model.init_from_base=True
    actor_rollout_ref.earl.training.tools=['calculator']
    algorithm.adv_estimator=trpo
    tool_config_path=tool_configs/combined_calculator.yaml
    data.train_files=./data/calc_bench/combined_earl/train.parquet
    data.val_files=./data/calc_bench/combined_earl/test.parquet
    data.train_batch_size=256
    data.max_prompt_length=384
    data.max_response_length=1024
    data.filter_overlong_prompts=True
    data.truncation='error'
    actor_rollout_ref.model.path=Qwen/Qwen2.5-7B-Instruct
    actor_rollout_ref.actor.optim.lr=1e-6
    actor_rollout_ref.model.use_remove_padding=True
    actor_rollout_ref.actor.ppo_mini_batch_size=32
    actor_rollout_ref.actor.ppo_micro_batch_size_per_gpu=8
    actor_rollout_ref.actor.use_kl_loss=True
```

```
1134        actor_rollout_ref.actor.kl_loss_coef=0.001
1135        actor_rollout_ref.actor.kl_loss_type=low_var_kl
1136        actor_rollout_ref.actor.entropy_coeff=0
1137        actor_rollout_ref.model.enable_gradient_checkpointing=True
1138        actor_rollout_ref.actor.entropy_checkpointing=True
1139        actor_rollout_ref.actor.entropy_from_logits_with_chunking=True
1140        actor_rollout_ref.actor.fsdp_config.param_offload=False
1141        actor_rollout_ref.actor.fsdp_config.optimizer_offload=False
1142        actor_rollout_ref.rollout.log_prob_micro_batch_size_per_gpu=8
1143        actor_rollout_ref.rollout.tensor_model_parallel_size=1
1144        actor_rollout_ref.rollout.name=vllm
1145        actor_rollout_ref.rollout.gpu_memory_utilization=0.6
1146        actor_rollout_ref.rollout.n=8
1147        actor_rollout_ref.ref.log_prob_micro_batch_size_per_gpu=32
1148        actor_rollout_ref.ref.fsdp_config.param_offload=True
1149        algorithm.use_kl_in_reward=False
1150        trainer.critic_warmup=0
1151        trainer.logger=['console','tensorboard']
1152        trainer.project_name='earl'
1153        trainer.experiment_name='calc_bench/qwen2.5-7b/earl-trpo'
1154        trainer.validation_data_dir=./val_result/calc_bench/7b/earl
1155        trainer.n_gpus_per_node=4
1156        trainer.nnodes=1
1157        trainer.save_freq=100
1158        trainer.test_freq=10
1159        trainer.total_epochs=4
```

## G.2 SORTING IMPLEMENTATION DETAILS

```
        ──────── Prompt+GRPO Training Config <Qwen-2.5-7B-Instruct> ────────
 apptainer exec --nv envs/verl.sif bash -c "
        ray start --head --port=6384 &&
        set -x &&
        python3 -m verl.trainer.main_earl
        actor_rollout_ref.earl.model.freeze_base_model=False
        actor_rollout_ref.earl.model.init_from_base=True
        actor_rollout_ref.earl.training.tools=['swap','compare']
        algorithm.adv_estimator=grpo
        tool_config_path=tool_configs/sorting_baseline.yaml
        data.train_files=./data/sort_baseline_train/train.parquet
        data.val_files=./data/sort_earl_4_5/test.parquet
        data.train_batch_size=256
        data.max_prompt_length=384
        data.max_response_length=384
        data.filter_overlong_prompts=True
        data.truncation='error'
        actor_rollout_ref.model.path=Qwen/Qwen2.5-7B-Instruct
        actor_rollout_ref.actor.optim.lr=1e-6
        actor_rollout_ref.model.use_remove_padding=True
        actor_rollout_ref.actor.ppo_mini_batch_size=32
        actor_rollout_ref.actor.ppo_micro_batch_size_per_gpu=8
        actor_rollout_ref.actor.use_kl_loss=True
        actor_rollout_ref.actor.kl_loss_coef=0.001
        actor_rollout_ref.actor.kl_loss_type=low_var_kl
        actor_rollout_ref.actor.entropy_coeff=0
        actor_rollout_ref.model.enable_gradient_checkpointing=True
        actor_rollout_ref.actor.entropy_checkpointing=True
        actor_rollout_ref.actor.entropy_from_logits_with_chunking=True
        actor_rollout_ref.actor.fsdp_config.param_offload=False
        actor_rollout_ref.actor.fsdp_config.optimizer_offload=False
        actor_rollout_ref.rollout.log_prob_micro_batch_size_per_gpu=8
        actor_rollout_ref.rollout.tensor_model_parallel_size=1
        actor_rollout_ref.rollout.name=vllm
```

```
1188        actor_rollout_ref.rollout.gpu_memory_utilization=0.6
1189        actor_rollout_ref.rollout.n=8
1190        actor_rollout_ref.ref.log_prob_micro_batch_size_per_gpu=8
1191        actor_rollout_ref.ref.fsdp_config.param_offload=True
1192        algorithm.use_kl_in_reward=False
1193        trainer.critic_warmup=0
1194        trainer.logger=['console','tensorboard']
1195        trainer.project_name='earl'
1196        trainer.experiment_name='sorting/qwen2.5-7b/prompt-grpo-sort'
1197        trainer.validation_data_dir=./val_result/sorting/7b/prompt-grpo
1198        trainer.n_gpus_per_node=4
1199        trainer.nnodes=1
1200        trainer.save_freq=5000
1201        trainer.test_freq=3
1202        trainer.total_epochs=1
```

```
                   ┌─ EARL Training Config <Qwen-2.5-7B-Instruct> ─┐
1203   apptainer exec --nv envs/verl.sif bash -c "
1204        ray start --head --port=6384 &&
1205        set -x &&
1206        python3 -m verl.trainer.main_earl
1207        actor_rollout_ref.earl.model.freeze_base_model=False
1208        actor_rollout_ref.earl.model.init_from_base=True
1209        actor_rollout_ref.earl.training.tools=['swap','compare']
1210        algorithm.adv_estimator=trpo
1211        tool_config_path=tool_configs/sorting.yaml
1212        data.train_files=./data/sort_earl_train/train.parquet
1213        data.val_files=./data/sort_earl_4_5/test.parquet
1214        data.train_batch_size=256
1215        data.max_prompt_length=384
1216        data.max_response_length=384
1217        data.filter_overlong_prompts=True
1218        data.truncation='error'
1219        actor_rollout_ref.model.path=Qwen/Qwen2.5-7B-Instruct
1220        actor_rollout_ref.actor.optim.lr=1e-6
1221        actor_rollout_ref.model.use_remove_padding=True
1222        actor_rollout_ref.actor.ppo_mini_batch_size=32
1223        actor_rollout_ref.actor.ppo_micro_batch_size_per_gpu=8
1224        actor_rollout_ref.actor.use_kl_loss=True
1225        actor_rollout_ref.actor.kl_loss_coef=0.001
1226        actor_rollout_ref.actor.kl_loss_type=low_var_kl
1227        actor_rollout_ref.actor.entropy_coeff=0
1228        actor_rollout_ref.model.enable_gradient_checkpointing=True
1229        actor_rollout_ref.actor.entropy_checkpointing=True
1230        actor_rollout_ref.actor.entropy_from_logits_with_chunking=True
1231        actor_rollout_ref.actor.fsdp_config.param_offload=False
1232        actor_rollout_ref.actor.fsdp_config.optimizer_offload=False
1233        actor_rollout_ref.rollout.log_prob_micro_batch_size_per_gpu=8
1234        actor_rollout_ref.rollout.tensor_model_parallel_size=1
1235        actor_rollout_ref.rollout.name=vllm
1236        actor_rollout_ref.rollout.gpu_memory_utilization=0.6
1237        actor_rollout_ref.rollout.n=8
1238        actor_rollout_ref.ref.log_prob_micro_batch_size_per_gpu=8
1239        actor_rollout_ref.ref.fsdp_config.param_offload=True
1240        algorithm.use_kl_in_reward=False
1241        trainer.critic_warmup=0
            trainer.logger=['console','tensorboard']
            trainer.project_name='earl'
            trainer.experiment_name='sorting/qwen2.5-7b/earl-sort'
            trainer.validation_data_dir=./val_result/sorting/7b/earl
            trainer.n_gpus_per_node=4
            trainer.nnodes=1
            trainer.save_freq=5000
```

```
trainer.test_freq=3
trainer.total_epochs=1
```

## G.3 PROMPTS

We highlight the environment prompt, question, and task in our instruction prompt.

```
──────── Prompt Example <Calc-Bench: Arithmetic> ────────
<system>:
You are a helpful assistant. You first thinks about the reasoning
process in the mind and then provides the user with the answer.

You are allowed to use calculator by wrapping the expression with
<calculator> </calculator> tags. The calculator output will follow
inside <result> </result> tags. For example,
<calculator> (12 + 6) / 3 </calculator> will produce
<result> 6 </result>. You can use calculator multiple times in
your reasoning process.

<user>:
What is 7 + (4 * 366 + 32287 - 7471)? Output your answer after '####'.

<assistant>:
Let me solve this step by step.
```

```
──────── Prompt Example <Calc-Bench: Countdown> ────────
<system>:
You are a helpful assistant. You first thinks about the reasoning
process in the mind and then provides the user with the answer.

You are allowed to use calculator with the ' calculate' keyword.
For example, you may calculate (12 + 6) / 3 = 6. You can use calculator
multiple times in your reasoning process.

<user>:
Using the numbers [3697, 5655, 1199, 11], create an equation that
equals 74587492. You can use basic arithmetic operations (+, -, *, /)
one or multiple times but each number can only be used once. Output the
final answer after '####'. For example, given numbers [1, 2, 3] and
target number 1, output #### (1 + 2) / 3.

<assistant>:
Let me solve this step by step.
```

```
──────── Prompt Example <Calc-Bench: Count> ────────
<system>:
You are a helpful assistant. You first thinks about the reasoning
process in the mind and then provides the user with the answer.

<user>:
How many times does the digit 2 appear in the number eighty-three
million, seven hundred and forty-five thousand and thirty-nine?
Output your answer after '####'. For example, given the number 121
and digit 1, output #### 2.

<assistant>:
Let me solve this step by step.
```

---

**Prompt Example <Calc−Bench: GSM8K*>**

**<system>:**
You are a helpful assistant. You first thinks about the reasoning
process in the mind and then provides the user with the answer.

**<user>:**
Natalia sold clips to 40770 of her friends in April, and then she sold
half as many clips in May. How many clips did Natalia sell altogether in
April and May? Output your final answer after '####'.

**<assistant>:**
Let me solve this step by step.

---

**Prompt Example <Sorting>**

**<system>:**
You are a helpful assistant. You first thinks about the reasoning
process in the mind and then provides the user with the answer.

You have access to the following tools:
− compare tool: compare A and B: A > B
− swap tool: swap A and B => B, A

**<user>:**
Sort the following items in descending order: A, B, C, D, E. While you
do not know the values of the items, you can compare any two items using
the compare tool. Once you know the order of them, you can use the swap
tool multiple times to complete the task. For example, to sort A, B in
descending order, if you find A < B with the compare tool, you can use
the swap tool on A and B to complete the task. Use fewest possible
comparisons and swaps to complete the task. Stop when the sequence is
sorted and do not output any answer.

**<assistant>:**
Let me solve this step by step.

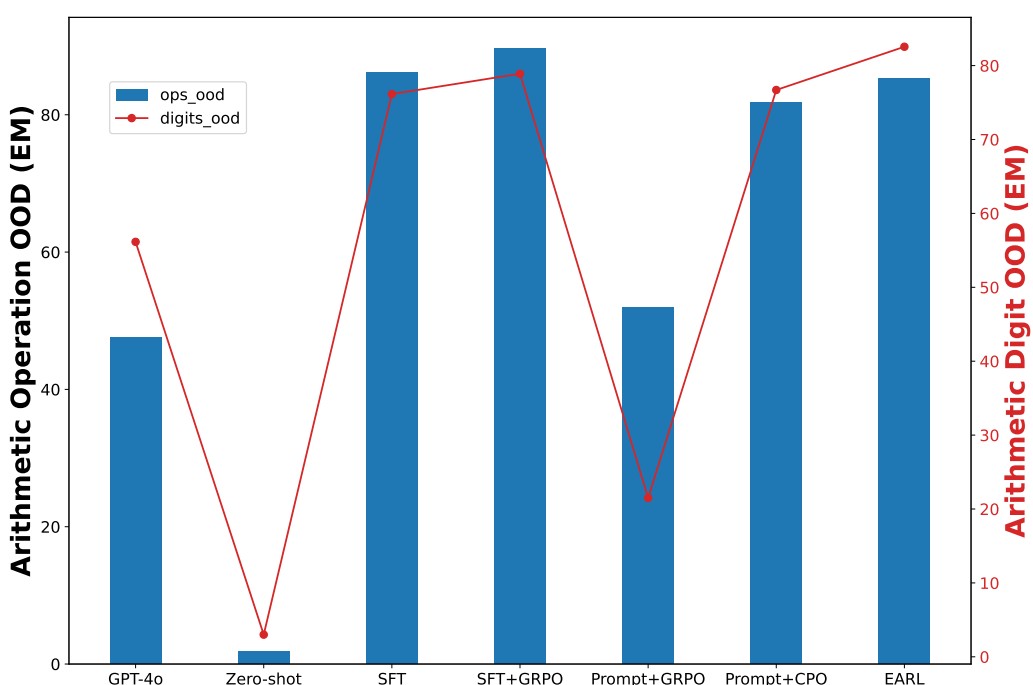

Figure 7: Performance comparison on the Calc-Bech Arithmetic. Arithmetic Operation Out-of-Distribution (OOD) is shown as bars, and Arithmetic Digit OOD as lines. Both plots correspond to the lightweight model Qwen-2.5-0.5B-Instruct.

Table 6: Statistics of the Arithmetic OOD setting. Language portions refer to the portion of questions where operations or numbers are written in natural language.

| Task | Max number ($10^x$) | | #Operands | | Lang. portion | | #Instances | |
|---|---|---|---|---|---|---|---|---|
| | Train | Test | Train | Test | Train | Test | Train | Test |
| Operation OOD | 4 | 4 | 3 | 6 | 5% | 70% | 20,000 | 2,000 |
| Digit OOD | 3 | 4 | 5 | 5 | NA | NA | 20,000 | 2,000 |

## H  ADDITIONAL EXPERIMENTAL RESULTS

### H.1  CALC-BENCH

**Evaluation on Lightweight Model.**   Experimental results in Figure 7 demonstrate that our EARL can effectively improve the Arithmetic task for a lightweight model with only 0.5 billion parameters (Qwen-2.5-0.5B-Instruct). In this experiment, as described in Table 6, we consider two Out-of-Distribution (OOD) variants, including *Operation OOD* and *Digit OOD*. Due to the limitation of the model's capability, we observe that Qwen-2.5-0.5B-Instruct has almost zero inherent knowledge to handle arithmetic tasks without fine-tuning. With enough training resources, SFT can achieve a significant performance gain for both OOD settings. Notably, EARL can effectively benefit the lightweight model, achieving comparable performance on the *Operation OOD* task, and even better results on the *Digit OOD* task.

**Evaluation on Collective Tool Learning.**   In Figure 8, we compare EARL with three baseline methods, where the tool learning trajectory of EARL is represented in yellow. We observed that all the methods in Figure 8d show a similar trend, demonstrating a distinct learning behavior with respect to other tasks. One possible reason is that this task is relatively easy, which could be well addressed by the model's inherent language knowledge spaces; introducing external action spaces won't enhance this task much. For the other three tasks, we observe that EARL reveals the efficiency and effectiveness of learning, indicating that all tasks are complementary to achieve enhanced and

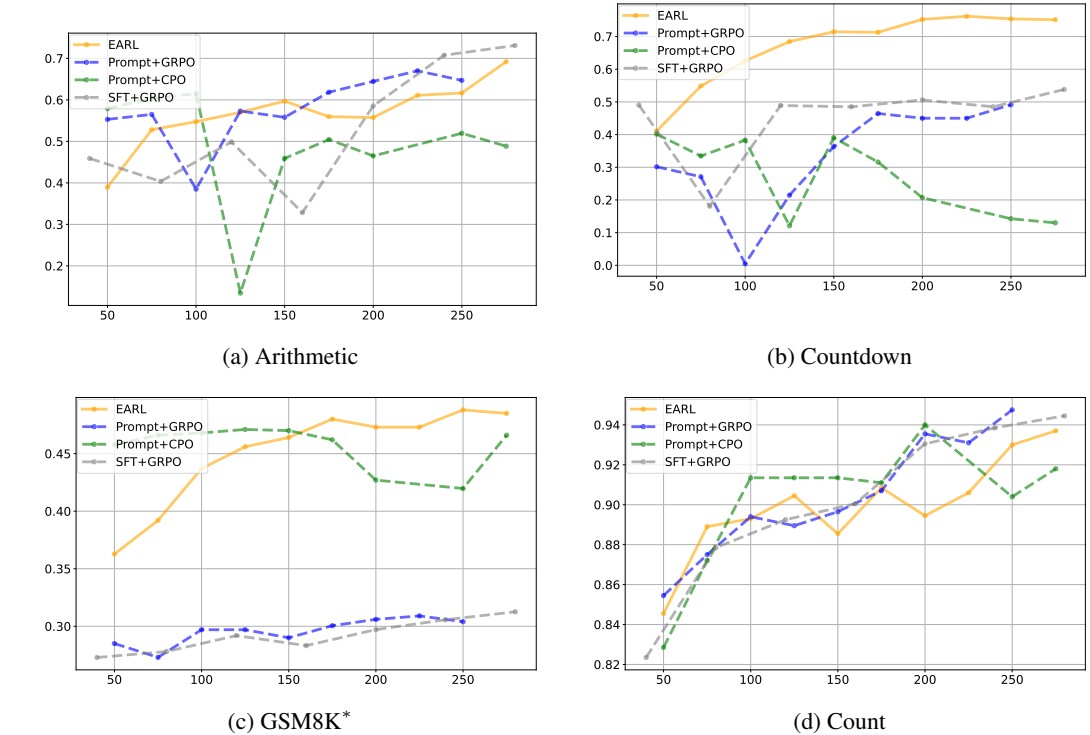

Figure 8: Performance of EARL (yellow) and other baselines on Calc-Bench, where we present the training steps on the X-axis and their EM score on the Y-axis.

Table 7: Ablation EM results of various environment configurations.

| Method | Configuration | Calc-Bench | | | | |
|---|---|---|---|---|---|---|
| | | **Arithmetic** | **Countdown** | **Count** | **GSM8K\*** | **Overall** |
| EARL | _calculate | 69.20 | 75.15 | 93.70 | 48.53 | 71.65 |
| | <calculator></..> | 75.80 | 78.00 | 92.00 | 51.12 | **74.23** |
| Prompt+GRPO | _<calculate>_ | 50.05 | 34.95 | 82.25 | 27.46 | 48.68 |
| | _<calculate> | 64.70 | 49.15 | 94.75 | 30.39 | 59.75 |
| Prompt+CPO | _<calculate>_ | 63.05 | 60.15 | 88.70 | 46.63 | 64.63 |
| | _<calculate> | 61.50 | 38.30 | 91.35 | 46.80 | 59.49 |

robust math reasoning with a better trade-off. Our findings suggest that EARL is a practical and scalable framework to expand action spaces.

**Additional Ablation Study.** To validate the robustness of EARL on different environment configurations, we conduct a comprehensive ablation study, shown in Table 7. All experimental results and findings validate that our proposed EARL achieves the best robustness, indicating that EARL is an effective framework with the potential to handle diverse external environments. Furthermore, by eliminating the need for environment-specific configurations, EARL develops a generalized understanding of tool interactions that is more naturally aligned with language.

## H.2 SORTING

Figure 9 visualizes the deterministic decision tree induced by EARL on Sort-4 inputs under greedy decoding. Each internal node represents a binary comparison, and each leaf corresponds to a sequence of swaps for producing a sorted output. Red nodes indicate redundant comparisons that do not affect the final decision and can be pruned to yield the simplified variant EARL* described in the main text.

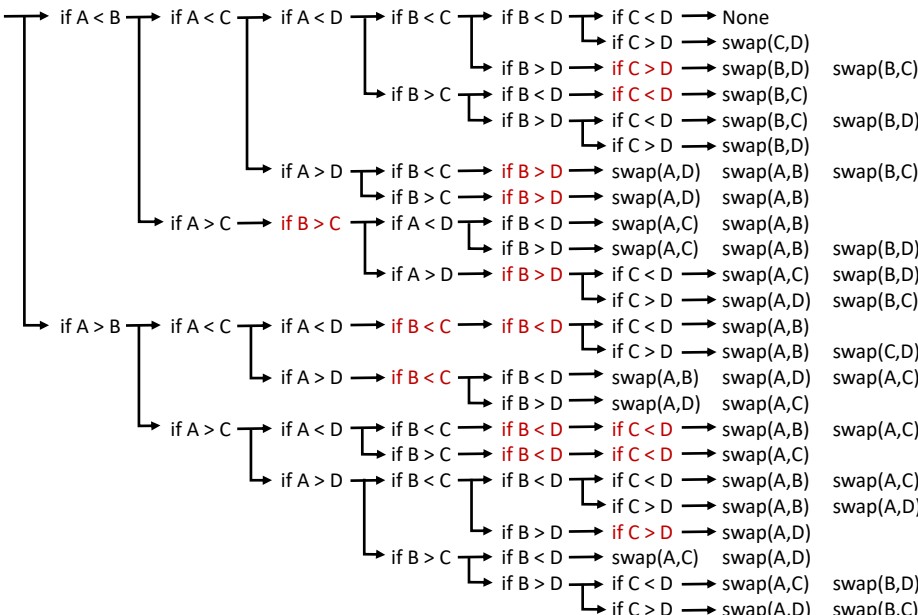

Figure 9: Decision tree induced by EARL on Sort-4. Red nodes indicate redundant comparisons that can be pruned to obtain EARL$^*$.

# I  ADDITIONAL ANALYSIS

## I.1  PLANNING PHRASES ANALYSIS

We have conducted an in-depth analysis of the contingent learning process on the Countdown task, where the intermediate response depends on the preceding actions and outcomes. The findings are summarized in Table 8, revealing distinct patterns across methods. For Prompt+GRPO, the dominant learning phrases are limited to 'different approach' and 'different combination', indicating a narrow exploration of planning strategies. In contrast, SFT+GRPO introduces greater diversity in planning phrase usage; however, the overall improvement in leveraging planning phrases is modest (6.4%), suggesting that supervised fine-tuning may introduce subjective biases that limit effective exploration.

Counterfactual Policy Optimization (CPO) markedly enhances planning diversity, increasing the number of planning phrases by 64.3% and 74.8% compared to SFT+GRPO and Prompt+GRPO, respectively. This demonstrates CPO's ability to enrich the planning process with a broader range of transition options expressed in natural language.

Notably, our proposed EARL achieves the highest utilization and diversity of planning phrases. EARL frequently employs conditional learning phrases such as 'not close', 'close to', and 'far from' when intermediate responses are incorrect, effectively guiding subsequent planning steps. This enables the model to either refine the current plan ('another combination') or initiate new strategies ('different combination' and 'different approach'), reflecting a more nuanced and adaptive planning behavior than other baselines.

## I.2  ACTION INITIALIZATION ANALYSIS

In Section 4.1 we described our policy parameterization and initialization strategy for expanded actions. Here we provide additional empirical evidence supporting this design. As shown in Figure 10, EARL-CPO with full initialization, rapidly learn to invoke tools and achieve the highest rewards. In contrast, models trained with EARL-CPO-no-init almost never use tools and instead converge to a suboptimal language-only strategy. We also evaluate a partial variant, EARL-CPO-init-step, in which only calculator button actions are initialized from their natural language descriptions, and find that it exhibits some tool use but remains less effective than full initialization in both tool utilization and reward.

Table 8: The number of using planning phrases across different training strategies.

| Phrase | EARL | Prompt+CPO | Prompt+GRPO | SFT+GRPO |
|---|---|---|---|---|
| not close | 9,961 | 99 | 0 | 18 |
| is close | 805 | 2,930 | 0 | 855 |
| close to | 3,019 | 1,138 | 0 | 2,253 |
| still close | 9 | 193 | 0 | 0 |
| different approach | 7,586 | 92 | 1,650 | 15 |
| another approach | 0 | 1,769 | 0 | 219 |
| different combination | 8,851 | 39 | 2,040 | 377 |
| another combination | 1,784 | 46 | 0 | 58 |
| negate | 682 | 4 | 0 | 121 |
| far from | 8,264 | 141 | 0 | 10 |
| **Total** | **40,961** | **6,451** | **3,690** | **3,926** |

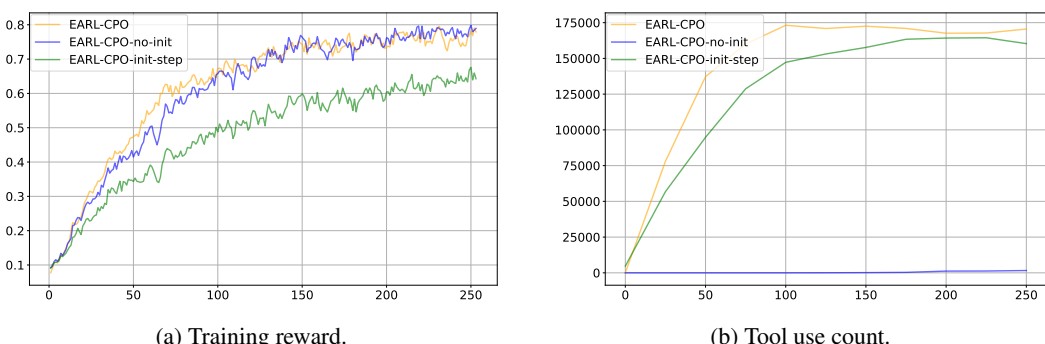

(a) Training reward.         (b) Tool use count.

Figure 10: Comparison of training reward and tool use count under different initialization strategies.

## J  CASE STUDY

Below are examples from the Countdown task under different methods.

```
┌───────────── EARL ─────────────┐
<question>:
Using the numbers [7, 681, 8, 6348], create an equation that equals
-5666.125.You can use basic arithmetic operations (+, -, *, /) one or
multiple times but each number can only be used once.
<EARL>:
We need to create an equation using the numbers 7, 681, 8,
and 6348 that equals -5666.125
One approach is to try different combinations of operations and see
if we can get close to the target.
Let's calculate 6348 - 681 - 7 - 8 = 5652
This is not close to -5666.125. Let's try another combination.
Let's calculate 6348 - 681 + 7 - 8 = 5666
We need to negate this to get -5666. Let's try negating it.
So, -(6348 - 681 + 7 - 8) = -5666
This is close, but not exactly -5666.125.
Let's try a different approach.
Let's calculate 6348 - 681 - 7 + 8 = 5668
This is not close either. Let's try a different combination.
Let's calculate 6348 - 681 + 7 + 8 = 5682
This is still not close. Let's try a different approach.
Let's calculate 6348 - 681 - 7 / 8 = 5666.125
We need to negate this to get -5666.125.
So, -(6348 - 681 - 7 / 8) = -5666.125
This gives us the correct result.
Therefore, the final answer is:#### -(6348 - 681 - 7 / 8)
```

```
<reward>:1.0

<question>:
Using the numbers [3, 8890, 6, 1], create an equation that equals
8888. You can use basic arithmetic operations (+, -, *, /) one or
multiple times but each number can only be used once.
<EARL>:
We need to create an equation using the numbers 3, 8890, 6, and 1
that equals 8888.
One approach is to try different combinations of operations
and see if we can get close to 8888.
Let's calculate 8890 - 8888 = 2
This isn't close to 8888. Let's try another combination.
Let's calculate 8890 - 6 - 1 + 3 = 8886
This is still not quite 8888. Let's try a different approach.
Let's calculate 8890 - 6 + 1 + 3 = 8888
This gives us the correct result.
Therefore, the equation is: 8890 - 6 + 1 + 3 = 8888
#### (8890 - 6 + 1 + 3)
<reward>:1.0
```

---
**Prompt+CPO**
---

```
<question>:
Using the numbers [7, 681, 8, 6348], create an equation that equals
-5666.125. You can use basic arithmetic operations (+, -, *, /) one or
multiple times but each number can only be used once.
<Prompt+CPO>:
First, I'll consider the numbers [7, 681, 8, 6348] and
try to form an equation that equals -5666.125.
One approach could be to use subtraction and division
to get close to -5666.125.
Let's try: <calculator> (6348 - 681) - 8 - 7 </calculator>
<result> 5652 </result> This is close but positive.
Another approach could be:
<calculator> (6348 - 681) - 8 + 7 </calculator>
<result> 5666 </result> This is close but positive.
Let's try: <calculator> (6348 - 681) + 8 - 7 </calculator>
<result> 5668 </result> This is close but positive.
Let's try: <calculator> (6348 - 681) + 8 + 7 </calculator>
<result> 5682 </result> This is close but positive.
Let's try: <calculator> (681 - 6348) - 8 - 7 </calculator>
<result> -5682 </result> This is close but positive.
Let's try: <calculator> (681 - 6348) - 8 + 7 </calculator>
<result> -5668 </result> This is close but positive.
Let's try: <calculator> (681 - 6348) + 8 - 7 </calculator>
<result> -5666 </result> This is the target number.
Therefore, the final answer is: #### (681 - 6348) + 8 - 7
<reward>:0.0

<question>:
Using the numbers [3, 8890, 6, 1], create an equation that equals
8888. You can use basic arithmetic operations (+, -, *, /) one or
multiple times but each number can only be used once.
<Prompt+CPO>:
First, I'll consider the numbers [3, 8890, 6, 1]
and try to form an equation that equals 8888.
One approach could be to use the largest numbers
first to get close to 8888.
Let's try: <calculator> (8890 - 6) - 3 - 1 </calculator>
<result> 8880 </result> This is close but slightly lower.
Another approach could be: <calculator> (8890 - 6) - 3 + 1 </calculator>
<result> 8882 </result>
This is also close but still slightly lower.
Let's try: <calculator> (8890 - 6) - 3 * 1 </calculator>
```

```
<result> 8881 </result> This is close but still slightly lower.
Let's try: <calculator> (8890 - 6) - 3 / 1 </calculator>
<result> 8881 </result> This is close but still slightly lower.
Let's try: <calculator> ( </calculator> <result> Error: '(' was
never closed (<string>, line 1) </result>
This approach seems to be incorrect.
<reward>:0.0
```

---

**Prompt+GRPO**

```
<question>:
Using the numbers [3, 8890, 6, 1], create an equation that equals
8888. You can use basic arithmetic operations (+, -, *, /) one or
multiple times but each number can only be used once.
<Prompt+GRPO>:
We need to create an equation using the numbers [3, 8890, 6, 1] that
equals 8888, using each number only once and employing basic
arithmetic operations.
Let's try different combinations:
1. <calculator>3 + 8890 + 6 + 1 </calculator> <result> 8900 </result>
(too high)
2. <calculator>3 + 8890 + 6 - 1 </calculator> <result> 8898 </result>
(too high)
3. <calculator>3 + 8890 + 6 * 1 </calculator> <result> 8899 </result>
(too high)
4. <calculator>3 + 8890 - 6 + 1 </calculator> <result> 8888 </result>
(correct)
So, the equation that equals 8888 is:
<calculator>3 + 8890 - 6 + 1 </calculator> <result> 8888 </result>
The final answer is:#### 3 + 8890 - 6 + 1
<reward>:1.0

<question>:
Using the numbers [27, 72, 511, 9], create an equation that equals
-403. You can use basic arithmetic operations (+, -, *, /) one or
multiple times but each number can only be used once.
<Prompt+GRPO>:
We need to create an equation using the numbers [27, 72, 511, 9] that
equals -403, using each number only once and employing basic
arithmetic operations.Let's try different combinations:
1. <calculator>27 + 72 + 511 + 9 </calculator> <result> 619 </result>
(too high)
2. <calculator>27 + 72 + 511 - 9 </calculator> <result> 601 </result>
(too high)
3. <calculator>27 + 72 + 511 * 9 </calculator> <result> 4698 </result>
(too high)
4. <calculator>27 + 72 - 511 + 9 </calculator> <result> -403 </result>
(correct)
So, the equation that equals -403 is:
<calculator>27 + 72 - 511 + 9 </calculator> <result> -403 </result>
The final answer is:#### 27 + 72 - 511 + 9
<reward>:1.0
```

---

**SFT+GRPO**

```
<question>:
Using the numbers [692, 8, 6, 6], create an equation that equals
66432. You can use basic arithmetic operations (+, -, *, /) one or
multiple times but each number can only be used once.
<SFT+GRPO>:
First, let's consider the target number 66432 and the numbers
we have: 692, 8, 6, and 6. We need to find a combination of
these numbers using basic arithmetic operations to reach 66432.
Let's start by exploring multiplication and division since
```

```
they can significantly change the magnitude of the numbers.
First, let's try multiplying 692 by 8:
<calculator> 692 * 8 </calculator> <result> 5536 </result> Next,
let's multiply 5536 by 6:
<calculator> 5536 * 6 </calculator> <result> 33216 </result>
Finally, let's multiply 33216 by 2 <result>
#### (692 *8 ) *6 *2 = 66432.
<reward>:1.0

<question>:
Using the numbers [54, 749, 1, 9], create an equation that equals
-793. You can use basic arithmetic operations (+, -, *, /) one or
multiple times but each number can only be used once.
<SFT+GRPO>:
First, let's try to combine the numbers using basic
arithmetic operations to get close to -793.
We have the numbers: 54, 749, 1, 9
Let's start by experimenting with different combinations and operations.
First, let's try subtracting 749 from 54:
<calculator> 54 - 749 </calculator> <result> -695 </result>
Next, let's subtract 9 from -695:
<calculator> 54 - 749 - 9 </calculator> <result> -704 </result>
Finally, let's subtract 1 to get -793:
<calculator> 54 - 749 - 9 - 1 </calculator> <result> -705 </result>
#### (54 -749 -9 ) -1
<reward>:0.0
```