# OpenReview forum: "Expanding the Action Space of LLMs to Reason Beyond Language"
_ICLR.cc/2026/Conference — Submitted to ICLR 2026_

### Official Review · Reviewer_sN8d · 2025-11-01

**Soundness:** 1
**Presentation:** 2
**Contribution:** 3
**Rating:** 4
**Confidence:** 4

**Summary:**

The paper proposes ExpA, a method to add additional non-language tokens to LLMs for environment interaction. It also introduces EARL, a reinforcement learning algorithm training LLM for better using ExpA. It conducts experiments comparing with standard prompting and GRPO fine-tuning methods, showing strong performance on some mathematical reasoning and sorting tasks.

**Strengths:**

+ ExpA is a good intuitive method to explore how LLM should interact with the environment. Instead of directly using natural language, the additional action tokens introduced by ExpA can ensure more accurate interactions.
+ EARL is a novel RL method for fine-tuning LLMs for ExpA. The counterfactual rollouts ensure efficient exploration of the newly-added action space compared to the standard GRPO.
+ Experiments show strong performance on Qwen-2.5 3/7B models compared to non-ExpA natural language methods and/or standard GRPO.

**Weaknesses:**

+ The benchmark in the evaluation is not very fascinating. The usage of additional calculator has far been explored by many works. The usage of LLM to sort sequences are also less interesting, as it can be easily solved by many traditional non-machine learning algorithms. It could be much more interesting if the author can show the application of ExpA in robotics tasks.
+ Countdown and GSM8k\* use extremely large numbers, which is naturally a good fit of ExpA since it can learn how to use external calculator. However, such kind of tasks are not generally the use case of LLMs (people can use simply DFS for Countdown, and the GSM8k\* is a modified version just to enhance the importance of a calculator). Thus, the chosen application does not show the strong potential in applications of LLMs of introducing additional action tokens. The same also holds for Arithmetic.
+ For Arithmetic and Count, other two Calc-Bench tasks, ExpA does not show very strong performance against standard SFT/prompt + GRPO methods.
+ Table 2 lacks a baseline of Prompt+GRPO (presented later in ablation only for Countdown). The current numbers look that Prompt+CPO is worse than Prompt+GRPO. It does not show clear advantage of CPO in general domain.
+ The paper lacks some baselines of introducing external action tokens (such as ToolkenGPT). It only compares with baselines without such action tokens.

**Questions:**

+ Is the model fine-tuned on each of the task, or there is only one model fine-tuned with mixture of the tasks?
+ Please see weakness.

---

> ### Author Response · Authors · 2025-11-20
> **Author Rebuttal**
>
> Thanks for reading and reviewing the paper. Please find our response below.
>
> ### W1: the selected benchmark is not fascinating.
> Thank you for raising this point. We strongly disagree and clarify the motivation behind our benchmark choices below:
> - **Our goal is to introduce and study a fundamental algorithmic paradigm, not to propose a specific end-application.**  ExpA and EARL change *how* LLMs interact with external environments---through non-linguistic routing actions and RL over an expanded action space. To evaluate such a core design change, we require *clean, controlled environments* that isolate the algorithmic factors we aim to study.
> - **The tasks used are far from trivial for LLM-based tool agents.**  Even frontier models like GPT-4o struggle with multi-step, contingent planning in Calc-Bench and Sorting. These tasks require dynamically switching between environments, interpreting partial observations, and deciding the correct next intervention.
> - **Sorting is a well-established challenge in RL and algorithm-discovery research.** As demonstrated by AlphaDev (Nature 2023) [1], discovering sorting behavior through learning is far from trivial. Our setting mirrors this challenge: the agent must *discover* how to sort through step-wise interactions by itself, not execute a pre-written algorithm or learn from imitation.
> - **We evaluate diverse scenarios.**  Calc-Bench tests numerical capabilities in a fully-observed environment under diverse emphasis (arithmetic, numerical understanding, mathematical problem solving, searching), while Sorting provides a partially observable MDP. These represent two complementary classes of decision-making for stress-test.
> - **Robotics and multimodal expansions are exciting but out of scope for this paper.** Applying ExpA to robotics or richer multimodal tools requires substantial system-level engineering. Our current contribution establishes the feasibility and benefits of the ExpA paradigm in environments suitable for isolating core ideas. We are actively exploring extensions to multimodal and embodied setups as future work.
> In summary, the benchmarks are chosen not for superficial novelty but because they are **appropriate, challenging, and diagnostic** for evaluating the core algorithmic contributions of ExpA and EARL.
> [1] Faster sorting algorithms discovered using deep reinforcement learning.
>
> ### W2: using large numbers makes ExpA naturally advantaged; and such tasks do not represent common LLM use cases.
> Our choice of large-number variants is deliberate and grounded in methodological reasons:
> - **The goal of the paper is to demonstrate when and why LLMs *need* external environments, not what they already do well.**  Standard benchmarks with small numbers allow LLMs to succeed through memorization or pattern recall. Larger-number variants expose the model’s limitations and create a setting where external tools meaningfully matter.
> - **Baselines have equal access to the calculator.** The advantage does not come from "letting ExpA use a calculator". All baselines also have access to the same tool.
> - **Countdown is not about computing large numbers.** It requires multi-step decision-making, switching between reasoning and interactions, and adapting based on intermediate results. It serves as diagnostic test bench for real-world environments.
> - **Sorting tests problem abstraction, not a real-world use case.**  Just as in algorithm-discovery research (e.g., AlphaDev), the goal is to study *generalizable problem-solving capabilities*, not to claim sorting itself is a real-world use case for LLMs.
> ### W3: ExpA does not outperform baselines on Arithmetic and Count.
> In the multitask setting of Table 2, models are trained on a combination of 4 tasks. The objective is *balanced performance across all 4 tasks*, not maximizing any single one. Baselines often overfit to the easier Calc-Bench subtasks (Arithmetic, Count) but collapse on GSM8K* or Countdown. ExpA+CPO, in contrast, maintains *consistently strong performance across all tasks*. Importantly, even on the subtasks where a baseline overfits, its advantage over ExpA is only marginal, while its overall performance is substantially worse.
>
> ### W4: No Prompt+GRPO baseline. Prompt+CPO is worse than Prompt+GRPO.
> We have the following clarifications:
> - We first point out that Table 2 *does* include the Prompt+GRPO baseline for all Calc-Bench tasks.
> - We did not claim CPO work in general domain. Instead, we show in Table 4 that it's the better optimization objective when expanding action space (ExpA+CPO > ExpA+GRPO).
> - We explain possible reasons that CPO does not work without ExpA in line 376-392. Essentially, counterfactual interventions provide more chances for agents to interact with environments, but this leads to more conflict between language reasoning and action learning. This conflict is resolved in ExpA due to the expanded action space, hence ExpA+CPO leads to the best performance.

---

> > ### Author Response · Authors · 2025-11-20
> > **Author Rebuttal (Continued)**
> >
> > ### W5: The paper lacks some baselines of introducing external action tokens (such as ToolkenGPT).
> > - We note that ToolkenGPT is an earlier reference and does not incorporate reinforcement learning, making direct comparison challenging. It is also non-trivial to adapt it to our setting for fair comparison. However, for completeness, we report that EARL achieves 53.71% accuracy on GSM8K*, substantially outperforming ToolkenGPT’s reported 33% accuracy on GSM8K-XL. This highlights the significant advancement offered by our approach.
> > - We also did not compare with methods that expand the *token space*. They do not fall under the same setting, as they require extensive supervised fine-tuning (see line 76-79).
> > ### Q: Is the model fine-tuned on each of the task, or there is only one model fine-tuned with mixture of the tasks?
> > We train each model jointly on all tasks under a multitask learning setting (Table 2 caption).

---

### Official Review · Reviewer_KtTa · 2025-11-01

**Soundness:** 3
**Presentation:** 3
**Contribution:** 2
**Rating:** 4
**Confidence:** 3

**Summary:**

The paper proposes ExpA—an expanded action space that adds routing and environment-specific actions to an LLM’s policy—and EARL, an RL recipe that encourages the LLM to use these actions. The agent begins in a language environment; it can route into a tool (e.g., a calculator), where decoding is constrained to that tool’s valid actions, receives observations back as text, and then routes back to language. This design explicitly decouples reasoning tokens from control actions, eliminating the need for hand-engineered string parsers that detect special tool blocks. Weight initialization ties new action logits to the embeddings of their textual descriptions, and CPO injects counterfactual “forced routing” at likely positions inferred from the pretrained next-token distribution. Empirically, on an Arithmetic task, Countdown, GSM8K*, and a count-based task, as well as a partially observed Sorting setup, the approach often outperforms vocabulary-only baselines (Prompt+GRPO, Prompt+CPO, SFT+GRPO).

**Strengths:**

* **Conceptual clarity & clean mechanism.** The MDP framing and Algorithm 1 make the control flow crisp: select a routing action, then act within the environment from an action set, append observations to the history, and return to language.
* **Well-motivated learning recipe.** The policy head covers both vocabulary and expanded actions but is environment-masked; new action logits are initialized from their description tokens, giving a strong prior, and counterfactual rollouts encourage invoking tools at plausible points.

**Weaknesses:**

I sort them in decreasing order of importance to my score. I am primarily concerned about the first two, which, if fully addressed, would encourage me to increase my score.
* **The source of gains is under-isolated.** Two plausible drivers are (i) cleaner, in-distribution inputs to the LLM while inside tools (as seen in the case study appendix), and (ii) action-space restriction (fewer invalid generations along the learning process). Adding appropriate baselines (e.g., language-only tool calls with subset-token decoding when inside <calculator>, or cleaning the format of tool calls after returning)—is missing and would provide significantly more substantial insights.
* **CPO vs GRPO effects are mixed across tasks/baselines.** While EARL (ExpA+CPO) is often best and converges quickly on Countdown, some tables and plots show baselines where GRPO rivals or beats CPO on subtasks.
* **“Removes reliance on external parsers” is slightly overstated.** Routing/dispatch is no longer string-parsed, which is a real advance; however, the system still needs parsers for the routing tokens and appends textual descriptions of actions to history for learning/context, so scaffolding remains.
* **Positioning vs prior “expanded-action” agents.** The paper contrasts ExpA with a token-only tool calling, but the related work discussion underplays earlier systems that already output non-linguistic control (e.g., VLA/robotic policies like RT-2 \[1\]). Tightening this comparison would prevent over-claiming novelty and clarify what is new.
* **Reproducibility details.** Appendix configs reference many flags without a compact table mapping them to semantics, or without clearly explaining which common configs are used across different methods and what the specific differences are.

\[1\] Brohan, A., Brown, N., Carbajal, J., Chebotar, Y., Chen, X., Choromanski, K., Ding, T., Driess, D., Dubey, A., Finn, C., Florence, P., Fu, C., Arenas, M. G., Gopalakrishnan, K., Han, K., Hausman, K., Herzog, A., Hsu, J., Ichter, B., . . . Zitkovich, B. (2023). RT-2: Vision-Language-Action Models Transfer Web Knowledge to Robotic Control. *ArXiv*. [https://arxiv.org/abs/2307.15818](https://arxiv.org/abs/2307.15818)

**Questions:**

See weaknesses.

---

> ### Author Response · Authors · 2025-11-20
> **Author Rebuttals**
>
> Thanks for reading and reviewing the paper. Please find our response below.
>
> ### W1: source of gains is under-isolated.
> Thank you for this thoughtful point. We address both potential explanations:
> - **Cleaner inputs to the LLMs in tools is not the reason.** In Appendix Table 7, we include an ablation where ExpA uses *exactly the same textual formats* as the prompt-based baselines (e.g., the same `<calculator>...</calculator>` conventions). ExpA even performs *better* with the baseline’s format (74.23 vs. 71.65), and we postulate that this is because the Qwen base model we used is accustomed to this format of tool calls.
> - **Action-space restriction is also controlled for.** All baselines use constrained decoding that limits generation to the allowed tokens inside tool-call tags, which is the same with the action space restriction in ExpA. We also clarify this in the table on top of this page. We will clarify this more explicitly in the paper to avoid confusion.
> ### W2: CPO vs GRPO effects are mixed across tasks/baselines.
> Thank you for the careful reading. The inconsistency arises from comparing two different experimental settings:
> - **Table 4 (single-task training on Countdown).** This is the ablation isolating GRPO vs. CPO. Here, ExpA+CPO **consistently and clearly outperforms** ExpA+GRPO in both convergence speed and final accuracy.
> - **Table 2 (multitask training across four tasks).** In this harder setting, the goal is not to maximize any single subtask but to achieve *balanced performance across all tasks*. Some GRPO-based baselines perform well on one task (e.g., arithmetic) but collapse on others (e.g., GSM8K*). CPO, by contrast, provides *strong performance across all tasks*, which is the key metric for multitask agents. Moreover, even when a baseline overfits to one subtask, its performance on that task is just slightly better than ours.
> We will clarify this distinction in the paper to avoid confusion.
> ### W3: "removing reliance on external parsers" may be slightly overstated.
> Thanks for the thoughtful comments. By *removing external parsers*, we do not mean eliminating all forms of scaffolding. Rather, there is an important distinction between types of scaffolding:
> - As you noted, routing and dispatch in ExpA are no longer triggered by detecting textual patterns, thereby removing reliance on a specially designed parser and on the model’s instruction-following ability. To clarify, there are *no routing tokens* that need to be parsed. Routing actions (e.g., entering the calculator) reside in the expanded action space and are selected by the policy, not inferred from generated text.
> - The textual descriptions appended to the history are simply the observation interface of the environment. They play the same role as standard environment outputs (e.g., "A > B") that LLMs naturally consume. This form of scaffolding is inherent to any environment involving LLMs in the loop, and does not require any model-side instruction-following or adherence to predefined tool-call syntax.
> ### W4: positioning vs prior “expanded-action” agents.
> Thank you for pointing out. We will clarify below.
> - RT-2 expands the *token space* by introducing discretized robotic-control tokens and trains the model via SFT to output sequences of these action tokens. In other words, it keeps a *token-only interface* and needs costly fine-tuning to represent these new action tokens.
> - We discussed similar works (multimodal LLMs) in line 76-79. We also discuss action adaptors and other work on non-token control interfaces in related work. We appreciate this pointer and will revise the related-work section to clearly acknowledge RT-2 and similar VLA agents.
> - Note that our approach expands the *action space*, not the *token space*. The routing and environment actions in ExpA are never directly appended to the input context to LLMs, and hence we do not need costly fine-tuning to represent them. This is a key point that motivates our approach.
> ### W5: reproducibility.
> Thanks for carefully checking the Appendix. We will clarify those flags in the open-source repository, and point to the repository in the published version of the paper.

---

### Official Review · Reviewer_h2rk · 2025-11-02

**Soundness:** 3
**Presentation:** 3
**Contribution:** 3
**Rating:** 6
**Confidence:** 4

**Summary:**

This paper introduces Expanded Action Space (ExpA), a novel framework for training language models to interact with external tools and environments by directly selecting actions, rather than issuing text-based commands. The authors contend that existing tool-using LLMs depend on special tokens and external parsers, which hinders end-to-end learning and optimization. ExpA instead offers a unified action space where the model can either produce language tokens or choose environment actions, allowing reinforcement learning to jointly optimize reasoning and tool-use behaviors. To support this, the authors propose EARL, a reinforcement learning algorithm that incorporates counterfactual policy optimization to help the model identify when tool usage is advantageous. Experimental results on arithmetic and sorting tasks demonstrate improved performance and the emergence of procedural strategies, which the authors argue showcases ExpA’s potential to develop more capable and principled LLM-based agents.

**Strengths:**

1. The paper introduces the Expanded Action Space (ExpA), allowing language models to interface with external environments by selecting environment-specific actions directly, rather than encoding tool use through special tokens and external parsers. This clean separation between language reasoning and environment control supports true end-to-end training.

2. The authors present EARL (ExpA Reinforcement Learning), which incorporates counterfactual policy optimization (CPO) to encourage exploration of infrequent but critical routing actions.

3. Across multi-step, contingent-planning tasks, EARL consistently outperforms strong baselines that restrict actions to the model’s vocabulary. Ablation studies highlight the effectiveness of CPO over GRPO and demonstrate that the approach is effective even with base models lacking supervised fine-tuning.

4. On the sorting task (Sort-4), EARL achieves 100% accuracy, with the resulting policy converging to a compact, near-optimal decision tree in terms of swaps and comparisons.

**Weaknesses:**

-The work is a natural extension of RL-formulated code execution agents (e.g., LEVER, RA-ABL, RCI, CodeAct), tool-use planners with policy heads, and counterfactual credit assignment in modular RL. The abstraction change (tokens → actions) is in line with past works ideas.

-comparisons against strong modern baselines (SFT+function calling, ReAct-style agents, grammar-based parsers) under identical budgets are lacking. Moreover, ablations study that isolate the effect of counterfactual routing is missing.

-The paper argues for an “RL-first” tool-use paradigm, but the experiments do not convincingly isolate RL’s necessity or superiority. The results show that performance improves under specific setups, yet there is no systematic comparison against modern SFT+RLHF tool frameworks or hybrid architectures. The counterfactual rollout idea is interesting, but the description and ablation depth are insufficient to confirm it is the primary driver versus simpler credit assignment heuristics.

-Moreover, environments used in experiments are toy-scale. Evaluation is limited to small, closed environments (arithmetic helpers, sorting tasks). These settings, while pedagogically clean, do not demonstrate scalability to heterogeneous real-world tools (APIs, OS environments, web search, memory systems, code interpreters). The method’s practicality in complex, partially observable, or noisy environments remains unclear. As a result, the claim of enabling “general tool-using agents” feels premature.

**Questions:**

-The paper claims to “remove the need for parsers” and introduce a fundamentally new paradigm, but this claim is weakened by the lack of engagement with prior work in RL-based tool agents, program-induction systems, and LLMs with environment-action heads (e.g., LEVER-like systems, code-exec agents, policy-head planning architectures, and work in embodied LLM agents). The contribution reads primarily as re-applying standard RL policy-head design to LLM tool use. The shift from textual tool invocation to direct action logits is a conceptual refinement, but the paper does not clearly differentiate itself from existing literature on action-space augmentation or structured decision heads in language-conditioned agents.  Can authors provide stronger comparative analysis to justify novelty.

-To strengthen the general-agent claim, the authors should evaluate the approach in richer, noisy, and heterogeneous tool settings rather than only toy arithmetic and sorting domains. In particular, it would be compelling to show that ExpA scales to real-world tools such as web search/browsing, OS/file manipulation, code execution, and persistent memory—and can operate across multiple tools within a single task.

-Experiments that incorporate partial observability, API failures, rate limits, and evolving tool schemas would demonstrate robustness and practical viability. Further, testing zero-shot generalization to unseen tools given only natural-language/tool-spec descriptions would clarify whether the architecture truly enables flexible tool use rather than overfitting predefined action templates.

Finally, comparisons against strong modern baselines (SFT+function calling, ReAct-style agents, grammar-based parsers) under identical budgets, and ablations isolating the effect of counterfactual routing, would help validate that the proposed action-space formulation—and not just engineering scaffolding—drives the observed performance gains.

---

> ### Author Response · Authors · 2025-11-20
> **Author Rebuttal**
>
> Thanks for reading and reviewing the paper. Please find our response below.
>
> ### W1 & Q1: the work does not clearly differentiate itself from existing literature.
> Thank you for the detailed comments. We respectfully disagree that ExpA is a natural extension of the cited works. The suggested literature uses *very different settings and approaches*. Below we clarify the distinctions concisely.
> - **LEVER / CodeAct**: These methods prompt LLMs to write code, and LEVER additionally trains a verifier to assess the code correctness. They do not expand the model’s action space beyond vocabulary. Agents are not fine-tuned for better interactions.
> - **RCI**: They prompt LLMs to operate computers. No expanded action space. No training to improve.
> - **RA-ABL**: We apologize for not being able to find the paper. Could you point to the paper you are referring to?
> - **Counterfactual credit assignment**: We did not find papers about "counterfactual credit assignment in modular RL". We assume you mean the counterfactual credit assignment paper in model-free RL. They use future-conditional (counterfactual) value functions as baselines or critics to provide low-variance training signals in RL. This is very different from our counterfactual policy optimization (CPO), whose goal is to encourage exploration of environments through counterfactual rollout.
> For further clarification, we highlight why our method provides a *fundamentally different way for agents to interact with external environments*:
> - **Decoupling reasoning from environment interaction.**  Prior works still require *language* to invoke a tool. ExpA introduces expanded, non-language actions for interactions, enabling true decoupling. This is not found in current coding or tool-use agents.
> - **No parsers or templates.**  Due to the above point, existing systems inevitably rely on parsing (`<calculator>...</calculator>`, JSON, code blocks, etc.). ExpA’s actions are direct policy outputs, not interpreted strings.
> - **CPO**.  Prior training paradigm relies on supervised fine-tuning or prompt-following capabilities of LLMs. Our CPO do neither of those and hence fully supports Zero-RL (RL on base models).
> ### W2 & Q4: comparisons against strong modern baselines under identical budgets are lacking.
> We disagree as *we include strong modern baselines* in the mainstream research, such as strong commercial-grade models (GPT-4o) and agentic LLMs trained with GRPO (without or with SFT). We provide further clarification below:
> - First, most our baselines (except GPT-4o) are essentially ReAct-style agents that interleave CoT reasoning, environment interactions and observations.
> - SFT+function calling corresponds to our SFT+GRPO baseline. SFT-only results are given in Figure 7 in appendix, which is worse than SFT+GRPO.
> - Most of our baselines (except GPT-4o) uses grammar-based parsers.
> - Identical training budget: Figure 8 in appendix provides performance comparisons vs. training steps, where our method still significantly outperforms baselines.
> ### W3: (summarized in bold below)
> - **The paper argues for "RL-first" but do not demonstrate it**: We did not argue for an "RL-first" paradigm, nor did we claim RL is necessary or superior. We want to design a new paradigm for LLMs to interact with environments that works with Zero-RL. Our finding is: *in our experimental settings*, RL, when combined with ExpA, consistently outperforms SFT-based approaches and prompt-based baselines.
> - **We do compare against modern SFT+RL hybrids.** Our SFT+GRPO corresponds to widely adopted SFT+RLHF tool-use pipelines. Across all tasks, ExpA+RL achieves substantially higher accuracy.
> - **Counterfactual rollouts vs. simpler credit assignment.**  Counterfactual rollout is to encourage exploration, not to improve credit assignment. The relevant baseline is other ways to encourage exploration, e.g., using prompts, which we incorporated in our experiment (Prompt+X) and our ExpA+CPO improves significantly over all of them. We also include a revised ablation table (modified from Table 4) on top of this page, demonstrating that CPO can further improve performance on top of prompting.

---

> > ### Author Response · Authors · 2025-11-20
> > **Author Rebuttal (Continued)**
> >
> > ### W4: toy-scale experiments, no heterogeneous real-world tools to demonstrate "general tool-using agents".
> > Thank you for raising this. We clarify our claims and the role of our chosen environments:
> > - **We never claim to build *general tool-using agents* in the paper.**
> > - **Our environments are intentionally controlled, not "toy" in the sense of being trivial.** The CalcBench and sorting tasks we evaluate are multi-step, possibly partially observable, and require contingent planning, which are properties shared with real-world tool interactions. In fact, even frontier models (e.g., GPT-4o) perform poorly on these settings, underscoring their non-trivial nature.
> > - **Demonstrating feasibility in clean settings is a necessary first step.**  Before scaling ExpA to heterogeneous, noisy, or OS-level tools, it is important to establish: (i) that non-language actions can be learned and invoked reliably, and (ii) that RL can optimize the expanded action space effectively. Our experiments show both, providing a foundation for future generalization.
> > - **Scalability to richer environments is a natural extension.**  Nothing in ExpA restricts it to arithmetic or sorting tools: new environments can be added in a modular fashion, and counterfactual exploration helps with initial learning when interactions are rare. Extending to web browsing, memory tools, or code interpreters is feasible (as discussed in our conclusion), but requires substantial engineering beyond the scope of this paper.
> > ### Q3: more settings (e.g., zero-shot generalization).
> > Thank you for these suggestions. We agree that what you mentioned are interesting directions, but they are well beyond the scope of the problem setting studied in this paper.
> > - **Zero-shot tool generalization is not the focus of our work**, and, to our knowledge, no existing tool-use or RL-based LLM agent framework (e.g., Toolformer, Retool, ARTIST) reliably achieves zero-shot operation of completely new tools from natural-language specifications alone. Prior work almost universally assumes: *predefined tool schemas, predefined tool interfaces, and/or SFT data demonstrating usage.* Our setting aligns with this standard evaluation protocol and our approach outperforms baselines.
> > - **Robustness to API failures, rate limits, or tool schema evolution** is certainly valuable. These topics typically require significant system-level engineering. While important, they are outside the scope of our conceptual and algorithmic contribution.
> > - **Our contribution focuses on a different challenge:** enabling LLMs to interact with tools *without requiring text parsing*, through a modular expanded action space.
> > Overall, future work could explore integrating ExpA with richer tool ecosystems, but our current results already demonstrate that the architecture is effective and competitive under the same evaluation standards used in the existing literature.

---

### Official Review · Reviewer_25KS · 2025-11-03

**Soundness:** 2
**Presentation:** 3
**Contribution:** 2
**Rating:** 4
**Confidence:** 3

**Summary:**

The paper proposes Expanded Action space (ExpA), a policy class where an LLM’s choices expand from vocabulary tokens to actions. The model can either stay in the language environment or route into an external environment and then select environment-specific actions until it routes back. To make models actually use these new actions, the authors introduce ExpA RL (EARL) with counterfactual policy optimization (CPO) trick. For each rollout, they build a counterfactual trajectory to encourage exploration of rarely invoked but crucial tools. On two benchmarks Calc-Bench and Sort-4, the approach outperforms prompt+RL and SFT+RL baselines.

**Strengths:**

1. A clean reframing of tool use as an explicit expanded action space, removing parser scaffolding.
2. The approach shows great improvement on the benchmarks (+10.46 EM overall), and it could have positive impacts on future LLM action capability and Agent development.
3. The paper is well written and easy to follow. It gives explanations of most of the technical details.

**Weaknesses:**

1. ExpA replaces an external parser with a learned action head, but observations are still appended as language tokens. This looks like a relocation of the parser boundary rather than a fundamentally different interface.
2. For large |E|, the performance of the approach is unclear. The author said the approach is a “scalable framework”, but this is a forward-looking claim. The paper doesn’t quantify behavior as the number of tools/actions grows.

**Questions:**

Most experiments are performed on Qwen 2.5 0.5, 3, 7b. Is the method still work on other models with different architectures or larger parameters?

---

> ### Author Response · Authors · 2025-11-20
> **Author Rebuttal**
>
> Thanks for reading and reviewing the paper. Please find our response below.
>
> ### W1: ExpA relocates the parser boundary rather than uses a fundamentally different interface.
> Thanks for raising this point. We respectfully disagree.
> - **Removal of parser creates a different interface.** In prior tool-use or agentic setups, the system must *interpret* the model’s natural-language output to detect tool calls (e.g., `<calculator>…</calculator>` or JSON patterns). This requires brittle, hand-crafted parsers and assumes the model can follow *precise instructions* on how to interact with environments (e.g., tool-call format). In ExpA, interactions with environments are selected directly from an expanded policy head (Section 4.1), not inferred from text. There is no parsing step between model output and environment execution.
> - **Textual observations are not part of parsing.** We only append observations from an environment, e.g., action description or function call result, which are defined by the environment itself. These are consumed by the model like any other text token. Crucially, these tokens are not parsed back into actions. They simply provide the information state of the environment. This is standard for any LLM agent interacting with text-based environments.
> Thus, the "parser boundary" is not relocated. It is removed from the agent's action-selection process entirely. What remains is the usual text-based observation interface inherent to LLM environments. We will clarify this distinction in the revision.
>
> ### W2: "scalable framework" is forward-looking.
> Thank you for pointing this out. We agree that our empirical results do not evaluate very large action sets. Our use of *scalable* refers to conceptual and architectural scalability. We will adjust the wording accordingly (e.g., "modular"). The key properties are:
> - **No supervised tool-call data or predefined language patterns are required:** Since ExpA does not rely on demonstrations or hand-crafted text templates, adding new environments or actions does *not* require collecting new supervised tool-call data or engineering new parsers. This avoids the main bottlenecks that limit the extensibility of parser-based agents.
> - **Modular integration of new environments.** Each environment contributes its own action set  $\mathcal{E}_i$, which is added directly to the expanded action head rather than being mediated through the language interface. Hence, adding a new environment does not interfere with existing ones or with the model’s language reasoning. This modularity is designed to scale more gracefully as $\mathcal{E}$ grows, since each environment is self-contained rather than competing for linguistic patterns or parser rules.
>
> ### Q:  Method still work on other models with different architectures or larger parameters?
> We first point out that evaluating across architectures and large model size is very computationally expensive. For this reason, we conducted comprehensive experiments on Qwen 2.5 family (0.5B, 3B, 7B).
> - Regarding **model size**, we observe *significant improvements* over baselines consistently as model size increases ($0.5B\to 3B \to 7B$), which already provides strong validation on ExpA's effectiveness w.r.t. model-size scaling. Further scaling is outside our current compute budget. For example, a *single* experiment on Qwen-32B needs *64 H100 GPUs* for 1.5 days (https://github.com/hkust-nlp/simpleRL-reason/issues/69).
> - Regarding **model architecture**, since ExpA operates only through the final action head and does not rely on architecture-specific components, the framework is model-agnostic by design. We are actively extending evaluations to additional architectures, and will ping you if we get results within the discussion period.

---

### Author Response · Authors · 2025-11-20
**Summary of Review**

We sincerely thank all reviewers for their constructive feedback and ACs for their coordination. Below we summarize the strengths, the common concerns across reviews (with brief clarifications), and the revisions/new experiments we will incorporate.
### Strengths
- **Clear conceptual reframing**: All reviewers highlighted that ExpA provides a clean and intuitive mechanism on how LLMs should interact with environments.
- **Effective learning method**: EARL and counterfactual rollouts were consistently recognized as well-motivated and beneficial for exploration (h2rk, KtTa, sN8d).
- **Strong empirical results**: Reviewers noted consistent improvements across multi-turn, contingent-planning tasks (25KS, h2rk, sN8d).
- **Clarity of presentation**: Several reviewers appreciated the paper’s clarity in writing and the clean MDP formulation (KtTa, 25KS).
### Common Concerns and Brief Clarifications
- **Benchmark too simple**. The chosen tasks are controlled but far from trivial (e.g., GPT-4o struggles significantly). Our goal is to study a fundamentally new algorithmic paradigm, and  the chosen benchmark is **appropriate, challenging, and diagnostic** for evaluating the core algorithmic contributions of ExpA and EARL.
- **Scaling to more complex applications / larger $|\mathcal{E}|$.**  Before scaling ExpA to more complex scenarios, it is important to show that non-language actions can be learned and invoked reliably. Our results strongly support this and we feel the contributions are already strong. Conceptually, ExpA allows modular integration of new environments, and hence scaling to larger $|\mathcal{E}|$ should be an advantageous setting for ExpA that we will demonstrate in future work.
- **Parser boundary relocation/remaining scaffolding**. We clarify that ExpA _removes_ parsing from action selection. Text appended to the history is simply environment observation (standard in prior work), and does not introduce the issues associated with parser-based action extraction (details in 25KS W1 and KtTa W3).
- **Baseline issues / related work**. We clarify that relevant baselines are not missing (h2rk), and comparisons with baselines are fair (KtTa). While the broader scheme of related work is in the paper, some specific lines of work can be added to improve clarity (KtTa).
### Planned Revisions & Added Experiments
- **Revisions**
    - Rewrite claims around "scalability" $\to$ "modular".
    - Expand comparisons to RT-2 and code-exec agents.
    - Clarify removing parser and the exact role of textual observations.
    - Clarify action restriction in environments is implemented for baselines as well for fair comparison.
    - Explain the setting differences between Table 2 and 4 clearer.
    - Clarify config for reproducibility in the open-source repository.
- **New Experiments**
    - We are adding experiments using a different model family.
---
We also reformat Table 4 for a clearer view of the differences between methods.

| Method             | SFT | Constrained decoding in env. | w/ env. prompt | Zero-RL | counterfacual rollout | ExpA | EM results |
| ------------------ | :-: | :--------------------------: | :------------: | :-----: | :-------------------: | :--: | :--------: |
| SFT-GRPO           |  √  |              √               |       √        |         |                       |      |   62.05    |
| Prompt-GRPO (Zero) |     |              √               |       √        |    √    |                       |      |   51.15    |
| Prompt-GRPO        |     |              √               |       √        |         |                       |      |   58.16    |
| **EARL**           |     |              √               |       √        |         |           √           |  √   |   80.09    |
| - prompt           |     |              √               |                |         |           √           |  √   |   76.76    |
| - CPO              |     |              √               |       √        |         |                       |  √   |   75.10    |
| - CPO & prompt     |     |              √               |                |         |                       |  √   |   73.79    |
| **EARL (Zero)**    |     |              √               |       √        |    √    |           √           |  √   |   77.31    |
| - prompt           |     |              √               |                |    √    |           √           |  √   |   74.56    |
| - CPO              |     |              √               |       √        |    √    |                       |  √   |   76.45    |
| - CPO & prompt     |     |              √               |                |    √    |                       |  √   |   70.27    |

**Highlighted observations**:
1. CPO encourages better environment interactions, even when environment prompts are available (e.g., EARL is better than EARL-CPO).
2. With ExpA, CPO is consistently better than GRPO.
3. All methods implement action space restriction when interacting with environments.

---

### Meta-Review · Area_Chair_XZUu · 2026-01-08

**Summary:**

This submission introduces an approach that expands the action space of Large Language Models (LLMs) beyond language, enabling explicit interaction with external environments. By decoupling environment interactions from language, the model can trigger actions in an expanded action space (ExpA) and receive feedback from external environments. The authors proposed ExpA Reinforcement Learning (EARL), which uses counterfactual policy optimization to improve to promote effective explorations of the expanded action space. Experimental results show that EARL outperforms strong baselines on multi-turn tasks, achieving significant improvements in tasks like calculator-based multi-task learning and partially observed sorting problems.

**Reviewer Concerns:**

The reviewers' concerns can be summarized into the following points:

1) Limited Evaluation Scope and Performance Comparison: The approach is evaluated on relatively simple tasks like sorting and arithmetic, which are not particularly novel and can be solved with traditional algorithms. Additionally, the paper lacks comprehensive comparisons with state-of-the-art baselines, such as Prompt+GRPO, and does not include necessary baselines like ToolkenGPT to highlight the benefit of introducing external action tokens.

2) Unclear Contribution of Counterfactual Optimization and Reward Design: The paper does not isolate the effects of counterfactual policy optimization (CPO) versus general reinforcement policy optimization (GRPO) across tasks, and the source of performance gains remains underexplored. The authors also fail to provide sufficient ablation studies to clarify the impact of their action-space framework and the possible advantages over simpler credit assignment methods.

3) Scalability and Practicality Concerns: While the method removes reliance on external parsers, it still requires token-based parsing for routing actions, which is seen as a shift in the parser boundary rather than a fundamental change. Furthermore, the experiments are limited to toy-scale environments, with no clear demonstration of scalability or applicability to real-world, complex, and noisy environments. The claim of enabling general tool-using agents is not convincingly supported by the current evaluation.

**Reviewer Scores:**

Four reviewers rated 4, 6, 4, 4 respectively. According to the authors' substantial rebuttal, AC thinks some reviewers may increase the scores slightly, given more explanations and re-presentation like Table 4. However, in terms of the scalability and practicality concerns, AC considers that the reviewers may not totally agree with the authors' feedback. To some extent, even you use a simple benchmark to validate the effectiveness of the proposed paradigm, it is not hard to verify in the less-perfect scenarios to justify the method. Therefore, AC consider that the concerns raised by the reviewers may not be sufficiently addressed.

---

### Decision · Program_Chairs · 2026-01-26

Reject